

# Integrable models on Rydberg atom chains

Luke Corcoran[1][*], Marius de Leeuw[1][†] and Balázs Pozsgay[2][‡]

1 School of Mathematics & Hamilton Mathematics Institute, Trinity College Dublin, Ireland
2 MTA-ELTE "Momentum" Integrable Quantum Dynamics Research Group,
Department of Theoretical Physics, ELTE Eötvös Loránd University, Budapest, Hungary

[*] lcorcoran@maths.tcd.ie , [†] mdeleeuw@maths.tcd.ie , [‡] pozsgay.balazs@ttk.elte.hu

## Abstract

We initiate a systematic study of integrable models for spin chains with constrained Hilbert spaces; we focus on spin-1/2 chains with the Rydberg constraint. We extend earlier results for medium-range spin chains to the constrained Hilbert space, and formulate an integrability condition. This enables us to construct new integrable models with fixed interaction ranges. We classify all time- and space-reflection symmetric integrable Rydberg-constrained Hamiltonians of range 3 and 4. At range 3, we find a single family of integrable Hamiltonians: the so-called RSOS quantum chains, which are related to the well-known RSOS models of Andrews, Baxter, and Forrester. At range 4 we find two families of models, the first of which is the constrained XXZ model. We also find a new family of models depending on a single coupling $z$. We provide evidence of two critical points related to the golden ratio $\phi$, at $z = \phi^{-1/2}$ and $z = \phi^{3/2}$. We also perform a partial classification of integrable Hamiltonians for range 5.

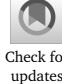

## Contents



# 1 Introduction and summary

Integrable spin chains are special many-body systems with distinctive features. They possess a typically infinite family of conserved charges, which constrain their dynamical processes, and often lead to exact solutions in various dynamical scenarios. It is an important task to classify integrable spin chains, and to understand the algebraic origin of the extra conservation laws.

The most well-studied integrable spin chains are those with nearest-neighbour interactions. Prototypical examples include the Heisenberg XXX spin chain [1] and the Hubbard model [2]. The key algebraic structure underlying the integrability of such models is the Yang–Baxter equation, discovered independently by Yang [3] and Baxter [4] in different contexts. A common algebraic framework for solving such models based on the Yang–Baxter equation has since been developed, known as the algebraic Bethe Ansatz [5].

There are fewer results for integrable spin chains in the case where the interaction range of the Hamiltonian is greater than two. The first distinguished class of such Hamiltonians are *long-range* models, whereby each spin can interact with any other spin on the chain. The formulation of integrability for such models is more involved than the nearest-neighbour case. Examples of long-range integrable spin chains include the Haldane-Shastry model [6,7] and the Inozemtsev model [8].

An even less-studied class of integrable Hamiltonians are those with a finite interaction range $r > 2$. An integrability framework for such models was proposed in [9], which dubbed these models *medium-range* in order to distinguish them from the nearest-neighbour and long-range case. While all integrable nearest-neighbour spin chains have higher-range commuting charges which could be interpreted as integrable Hamiltonians generating higher-range dynamics, we reserve the denomination of medium-range for cases where the Hamiltonian does not arise as a such a higher charge. Such models are relatively sparse in the literature; examples include the Bariev model [10], the folded XXZ model [11–14], and the range 3 model of Fendley [15].

Recently there has been a surge in interest in *classifying* integrable spin chains in various scenarios, which in many cases can equivalently be stated as solving the Yang–Baxter equation. Key ingredients in many such approaches are the Reshetikhin condition [16], the boost

operator [17, 18], and the Sutherland equations [19]. These have been combined to fully classify integrable spin chains with a two-dimensional local state space, which can be derived from regular solutions of the Yang–Baxter equation [20–25]. These works also include partial results for the case of local Hilbert space dimensions three and four. We mention that similar methods have been applied in other many other works to construct new integrable spin chains and solutions of the Yang–Baxter equation in different setups [26–31]. More recent works include [32–40]. There are less classification results in the case of medium-range integrable models, although an initial step was made in [9], see also [41, 42].

In all cases considered above the quantum spin chains were defined on a Hilbert space with a standard tensor product structure. Such models do not necessarily exhaust all possibilities for integrability. In fact, it has been known for a long time that there exist integrable models defined on *constrained Hilbert spaces*. In such cases there is a local constraint for the allowed basis states in the Hilbert space, and the Hamiltonian preserves this constraint. The models in question are integrable within the constrained Hilbert space, and it is generally not guaranteed that there is a corresponding integrable model acting on the full Hilbert space, whose restriction would give the constrained integrable model. Therefore, the treatment of such models is somewhat different from the standard cases.

An important class of local constraints is when in the computational basis only certain pairs of one-site states are allowed at neighbouring positions. A famous example is the so-called *Rydberg blockade constraint*, which arises in experimental situations with Rydberg atom chains [43]. In this case the Hilbert space consists of those states of a spin-1/2 chain, where there are no down spins allowed on neighbouring sites.[1] There are no interesting nearest-neighbour Hamiltonians on Hilbert spaces with such a constraint. Therefore one must consider either medium-range or long-range models.

It has been known for a long time [44] that this constraint can be interpreted as a special case of the so-called Restricted Solid On Solid (RSOS) models of Andrews, Baxter and Forrester [45, 46]. The RSOS models are two dimensional statistical physical models defined on a square lattice, with dynamical variables at the vertices (taking values $1, \dots, \ell$) and statistical weights associated with the plaquettes. Only those classical configurations are allowed, where the variables on neighbouring vertices differ precisely by $\pm 1$. Quantum spin chain Hamiltonians associated with the RSOS models were studied in [44]. The RSOS models serve as lattice discretisations of minimal models of CFT [47, 48] and also off-critical QFTs [49, 50].

Quantum spin chains with the Rydberg constraint were studied in [51]. This work considered a two-parameter range 3 Hamiltonian on the constrained Hilbert space. Although this model is not integrable for generic parameter values, there is a one-dimensional integrable line where the model coincides with the integrable family derived from the RSOS models [44]. A specific point for this family of models is the so-called golden chain [52]: it is a critical spin chain, leading to two different CFTs in the vicinity of the ground state and the anti-ground state. The golden chain is related to a fusion category of non-abelian anyons [53]. A closely related but non-integrable one-parameter family of Hamiltonians was considered by Lesanovsky [54, 55], see also [56].

Another known family of integrable models with the Rydberg constraint is the so-called constrained XXZ model [57–61]. In this case the Hamiltonian has range 4 interactions, and the models are Bethe Ansatz solvable. The generalised hydrodynamics for this model has been constructed in [62]. A special point for this family (with coupling $\Delta = 0$) arises as a strong coupling limit of the XXZ spin chain [63], and in this case the model is embedded into the folded XXZ model [11–14], which is defined on the full Hilbert space.

---

[1]The definition depends on the conventions chosen. In the actual experimental setup it is not possible to have excited Rydberg states on nearby atoms [43]. We represent such excited states by a down spin embedded into a sea of up spins, and this explains our description for the constraint.

Recently there has been considerable interest in models with the Rydberg constraint, motivated by the experimental finding of [64], observing the lack of thermalisation in a Rydberg chain. This led to the discovery of quantum many-body scars in such systems, more specifically in the so-called PXP model [65, 66] (see also the reviews [67, 68]). It is a natural question, whether the exotic properties of the PXP are related in any way to integrability, perhaps to integrability of the model itself or of models "nearby" [69, 70]. Now it is generally believed that the PXP model is not integrable, and it was proven rigorously in [71] that it does not possess conserved charges with local densities. Nevertheless the question remained, whether there can be integrable models on the constrained Hilbert space, perhaps with longer interaction ranges, which are in some sense close to the PXP model. We note that alternative explanations for the many-body scarring have been put forward, including a non-integrable parent model which supports an emergent $SU(2)$ spin dynamics in a subspace of the Hilbert space [72, 73].

We should mention that models defined on Rydberg chains display anomalous transport properties [74, 75]. And there is ongoing experimental work to realize such spin chains; for the constrained XXZ chain see for example [76, 77].

In this work we extend the methods of integrability to find and to classify models on constrained Hilbert spaces, focusing on the case of the Rydberg blockade. We formulate an algebraic criterion for a model to be integrable on the Rydberg-constrained Hilbert space. With this criterion it is straightforward to show that the PXP model is not Yang-Baxter integrable, and allows us to analytically find new integrable models on the constrained Hilbert space. We note that some other criterions for integrability, for example Poissonian level statistics [70], only allow for a numerical test of integrability.

Our aim is to construct integrable models with higher interaction ranges. There are several potential applications for new integrable models on the constrained Hilbert space: any such model is a candidate for the potential "parent integrable model" proximate to the PXP model [70]. There is also the potential for interesting algebraic structures and the appearance of CFTs in the large $L$ limit.

In order to make this paper accessible to the non-specialist reader, below we summarise our key findings. In particular, we list the integrable models that we find for interaction ranges 3 and 4.

## 1.1 Summary

We briefly summarise the logic and results of our paper. In section 2 we review known results for integrable medium-range spin chains [9]. One of the key results we will adapt is a criterion for integrability at the level of Hamiltonian $Q_2$. This criterion is analogous to the Reshetikhin condition [16] for nearest-neighbour spin chains and takes the form of

$$[Q_2, Q_3] = 0, \tag{1}$$

where $Q_3$ is an operator of higher range than the Hamiltonian.[2] Crucially, this operator can be constructed directly from the Hamiltonian using a higher-range analogue of the so-called boost operator [17, 18]:

$$Q_3 = \mathcal{B}(Q_2), \tag{2}$$

and so (1) can be formulated as an integrability condition for $Q_2$ directly. If a Hamiltonian satisfies (1) then it can be related to a Lax operator $\mathcal{L}$, which satisfies the so-called RLL relation. $R$ is the $R$-matrix, which satisfies the Yang-Baxter equation. Using the RLL relation one can

---

[2]The indexing $Q_\alpha$ of the conserved charges follows from earlier conventions, where $Q_2$ was chosen as the Hamiltonian of range $r$, and $Q_3$ was a higher charge of range $2r - 1$. $Q_1$ was a charge generating translations along the spin chain.

construct an infinite tower of commuting charges $Q_j$, a key feature of quantum integrable systems.

In section 3 we discuss models on Rydberg-constrained Hilbert spaces. Such Hilbert spaces can be obtained from a factorised Hilbert space $\bigotimes_{i=1}^{L} \mathbb{C}^2$ by the insertion of a projection operator $\Pi$, which projects out states with neighbouring down spins. Such a projection destroys the product form of the Hilbert space, which the RLL formulation of integrability relies on. In section 4 we examine how to adapt this formulation of integrability to a non-factorised Hilbert space. The condition (1) can be adapted by simply inserting projectors:

$$\Pi\left[Q_2, Q_3\right]\Pi = 0, \tag{3}$$

which allows for a systematic way to search for integrable Hamiltonians on the constrained Hilbert space. The RLL relation and the Yang-Baxter equation are inherently local equations, however, and must be modified appropriately in order to prove integrability of a model satisfying (3). We find that the Lax operator and $R$-matrix can be modified $\mathcal{L} \to \tilde{\mathcal{L}}, R \to \tilde{R}$ by inserting appropriate local projectors on their auxiliary space indices. In this way, we propose projected versions of the RLL relation and Yang–Baxter equation which an integrable constrained model should satisfy.

We also discuss the so-called GLL formulation of integrability, introduced in [9], on the constrained Hilbert space. This formalism can be applied when the Lax operator takes a special form, namely when it acts diagonally on the first and last site. We argue that Lax operators for integrable constrained models can always be written in this form, and so the GLL formulation of integrability is natural. The $G$-operator contains the same information as the $R$-matrix, but acts on one less site. The $G$-operator is also the object which exposes the correspondence between one-dimensional quantum Hamiltonians acting on constrained Hilbert spaces, and two-dimensional statistical physics models on a square lattice. In section 5 we review this correspondence for range 3 models, and extend it to the range 4 case.

In sections 6, 7, and appendix A, we analyse the equation (3) for Hamiltonians of range 3, 4, and 5. We focus on the case of time- and space-reflection invariant models. For range 3 and range 4 we can solve (3) exactly, and hence fully classify all such integrable models on the Rydberg-constrained Hilbert space. At range 3 we find a single family of integrable Hamiltonians, which we parametrise as

$$\mathcal{H}_{123} = P_1 X_2 P_3 + z P_1 P_2 P_3 + \left(2z + \frac{1}{z}\right) P_1 N_2 P_3, \tag{4}$$

where $X$ is the first Pauli matrix, and $P$ and $N$ are operators projecting locally onto up and down spins respectively. This model is well-known [44,51], and is related to the RSOS models of Andrews, Baxter, and Forrester [45,46]. This model provides a useful verification of our formalism. The family has a single critical point at $z = \phi^{5/2}$, which corresponds to the so-called golden chain [52].

At range 4 we find two families of models. The first one is the constrained XXZ model [57,58]:

$$\mathcal{H}_{1234}^{\text{XXZ}} = P_1\left(\frac{X_2 X_3 + Y_2 Y_3}{2}\right)P_4 + a P_1 P_2 P_3, \tag{5}$$

and the second is a new model:

$$\mathcal{H}_{1234}^{\text{DGC}} = P_1\left(\frac{X_2 X_3 + Y_2 Y_3}{2} + z(X_2 P_3 + P_2 X_3) + \frac{2z^2}{1-z^2}P_2 P_3\right)P_4 \tag{6}$$

$$+ P_1\left(-\frac{1+z^2}{z}X_2 + (z^2-1)P_2 + \frac{1-7z^4+2z^6}{z^2(z^2-1)}N_2\right)P_3.$$

Although (6) may look exotic, it is an integrable Rydberg-constrained Hamiltonian with interesting properties. Using a gap analysis we find numerical evidence that this model has critical points at $z = \phi^{-1/2}$ and $z = \phi^{3/2}$, where $\phi$ is the golden ratio. Because of this, we find it appropriate to call this model the double golden chain. At the critical point $z = \phi^{3/2}$, we show that the model (6) satisfies a Temperley-Lieb algebra in a non-standard way. We were unable to identify a similar algebraic structure at the point $z = \phi^{-1/2}$. We give expressions for the projected Lax operators of these models, and discuss their integrability properties. Finally, we provide a partial classification of integrable Rydberg-constrained Hamiltonians of range 5.

Returning to the non-integrable PXP model and the questions raised in [70], our key finding is that at interaction range 4 there is no exactly integrable model "close" to it, and even our partial classification at range 5 did not find any proximate integrable model.

## 2 Medium-range integrability

In this section we review the notion of integrability for medium-range spin chains. We give only necessary formulas and refer to [9] for full details. We consider translationally invariant Hamiltonians with periodic boundary conditions and a finite interaction range $r \geq 3$. We take the Hilbert space to be $V = \bigotimes_{j=1}^{L} V_j$, where each $V_j \simeq \mathbb{C}^d$ for a fixed $d \geq 2$. Due to translational invariance, we can realise the total Hamiltonian $Q_2 : V \to V$ as

$$Q_2 = \sum_{i=1}^{L} \mathcal{H}_{i,i+1,\dots,i+r-1}, \tag{7}$$

where $\mathcal{H} : (\mathbb{C}^d)^{\otimes r} \to (\mathbb{C}^d)^{\otimes r}$ is the Hamiltonian density. Translational invariance can be summarised by the commutation relation

$$[U, Q_2] = 0, \tag{8}$$

where $U : V \to V$ is the shift operator

$$U = \mathcal{P}_{12} \mathcal{P}_{23} \cdots \mathcal{P}_{L-1,L}, \tag{9}$$

and $\mathcal{P}_{ij}$ is the permutation operator on $V_i \otimes V_j$. One definition for integrability is the existence of infinitely many operators $Q_3, Q_4, \dots$ such that the whole set of operators $Q_j : V \to V$ mutually commute[3]

$$[Q_i, Q_j] = 0. \tag{10}$$

Due to this commutation, we will also refer to the $Q_i$ as charges. Importantly, we take $Q_2$ to be the operator of lowest range in this tower of charges. In particular, this excludes the possibility of $Q_2$ arising as a higher charge of an integrable nearest-neighbour model. For translationally-invariant Hamiltonians we can take the first charge $Q_1$ to be a generalised momentum operator. The shift operator is related to this via exponentiation $e^{Q_1} \sim U^{r-1}$.

**Lax operator and $R$-matrix.** Such a tower of commuting charges can be constructed if $\mathcal{H}$ can be obtained from a *Lax operator* $\mathcal{L}_{Ai}(u) : V_A \otimes V_i \to V_A \otimes V_i$. Here $u \in \mathbb{C}$ is a spectral parameter and $V_A \simeq (\mathbb{C}^d)^{\otimes(r-1)}$ is an auxiliary space whose dimension depends on the range $r$ of $Q_2$. We mention that for nearest-neighbour chains we have $r = 2$ and the dimensions of the auxiliary space and the local physical space coincide; the expansion of the auxiliary space is a new feature for higher-range models. This Lax operator generates charges for an integrable

---

[3]Note that the index on $Q_j$ refers not to its interaction range, but to its position in the tower of charges.

spin chain if there exists an invertible *R-matrix* $R_{AB}(u,v) : V_A \otimes V_B \to V_A \otimes V_B$ such that the *RLL relation* is satisfied

$$R_{AB}(u,v)\mathcal{L}_{Ai}(u)\mathcal{L}_{Bi}(v) = \mathcal{L}_{Bi}(v)\mathcal{L}_{Ai}(u)R_{AB}(u,v), \tag{11}$$

and $R_{AB}(u,v)$ satisfies the Yang–Baxter equation

$$R_{AB}(u,v)R_{AC}(u,w)R_{BC}(v,w) = R_{BC}(v,w)R_{AC}(u,w)R_{AB}(u,v). \tag{12}$$

For many models of physical relevance, the *R*-matrix satisfies the *regularity* property

$$R_{AB}(u,u) \propto \mathcal{P}_{AB}, \tag{13}$$

in which case the braiding unitarity condition is satisfied

$$R_{AB}(u,v)R_{BA}(v,u) \propto 1. \tag{14}$$

From the Lax operator one can form a homogeneous monodromy $T_A(u) : V_A \otimes V \to V_A \otimes V$ via

$$T_A(u) = \mathcal{L}_{AL}(u)\mathcal{L}_{A,L-1}(u)\cdots\mathcal{L}_{A1}(u). \tag{15}$$

The RLL relation (11) ensures that this operator satisfies the so-called RTT relation

$$R_{AB}(u,v)T_A(u)T_B(v) = T_B(v)T_A(u)R_{AB}(u,v). \tag{16}$$

From the monodromy matrix one can form a transfer matrix $t(u) : V \to V$ by tracing over the auxiliary space

$$t(u) = \mathrm{tr}_A T_A(u), \tag{17}$$

and from (16) it can be shown that

$$[t(u), t(v)] = 0. \tag{18}$$

The local charges of an integrable spin chain can then be obtained via logarithmic derivatives of the transfer matrix

$$Q_j = \frac{d^{j-1}}{du^{j-1}}\log t(u)\Big|_{u=0}. \tag{19}$$

For example, the first two charges can be computed as

$$Q_2 = \frac{d}{du}\log t(u)\Big|_{u=0} = t^{-1}(0)t'(0), \tag{20}$$

$$Q_3 = \frac{d^2}{du^2}\log t(u)\Big|_{u=0} = t^{-1}(0)t''(0) - (t^{-1}(0)t'(0))^2. \tag{21}$$

The commutation of the transfer matrix at different values of the spectral parameter (18) ensures that the full set of charges $Q_j$ mutually commute. The Hamiltonian density corresponding to the integrable Hamiltonian $Q_2$ can be obtained directly from the Lax operator

$$\mathcal{H}_{12\dots r} = \partial_u \check{\mathcal{L}}_{12\dots r}(u)\Big|_{u=0}, \tag{22}$$

where $\check{\mathcal{L}}_{12\dots r}(u) := \mathcal{P}_{r-1,r}\mathcal{P}_{r-2,r}\cdots\mathcal{P}_{1,r}\mathcal{L}_{12\dots r}(u)$.

**Higher charges from the Hamiltonian.** We call a range-$r$ Hamiltonian density $\mathcal{H}_{12\ldots r}$ integrable if it can be derived via (22) from a Lax operator satisfying the RLL relation (11) for some $R$-matrix $R_{AB}(u,v)$. In this case one can construct the higher charges $Q_3, Q_4, \ldots$ from the transfer matrix $t(u)$. In fact, for regular models there is a way to obtain the higher charge $Q_3$ directly from the Hamiltonian density. This is a higher-range analogue of the 'boost operator' approach for generating higher charges in integrable nearest-neighbour models [17, 18]. If the charge $Q_2$ has range $r$ then $Q_3$ has range $2r - 1$. By translational invariance we can write $Q_3$ as a sum over densities

$$Q_3 = \sum_{i=1}^{L} \mathcal{Q}_{i,i+1,\ldots,i+2r-2} \, . \tag{23}$$

The operator density $\mathcal{Q}$ can be obtained directly from $\mathcal{H}$ via

$$\mathcal{Q}_{12\ldots 2r-1} = [\mathcal{H}_{12\ldots r}, \mathcal{H}_{23\ldots r+1} + \ldots + \mathcal{H}_{r,r+1,\ldots,2r-1}] + q_{12\cdots r} \, , \tag{24}$$

for some range $r$ operator $q_{12\ldots r}$. In fact, $q_{12\ldots r}$ is related to the Hamiltonian density $\mathcal{H}$ and derivatives of the Lax operator $\mathcal{L}(u)$ via

$$q_{12\ldots r} \sim (\mathcal{H}_{12\ldots r})^2 - \partial_u^2 \check{\mathcal{L}}_{12\cdots r}\Big|_{u=0} \, , \tag{25}$$

but this explicit expression will not be necessary in this paper. Given $\mathcal{H}$, we can construct $\mathcal{Q}$ as a function of the undetermined range $r$ operator $q_{12\ldots r}$ using (24). Then a necessary condition for integrability is that there exists a choice of $q_{12\ldots r}$ such that[4]

$$[Q_2, Q_3] = 0 \, . \tag{26}$$

In fact, the condition (26) appears to be a sufficient condition for integrability. Although a proof of this fact is so far missing, all known Hamiltonian densities $\mathcal{H}$ satisfying (26) have proven to be integrable. This leads to an approach for generating and potentially classifying integrable Hamiltonians $\mathcal{H}$:

- Parametrise a general Hamiltonian density $\mathcal{H}_{12\ldots r}$ for some range $r$ and local Hilbert space dimension $d$, potentially requiring for some symmetries. Compute the full charge $Q_2$.

- Form the higher charge density $\mathcal{Q}_{12\ldots 2r-1}$ in terms of an undetermined range $r$ operator $q_{12\ldots r}$ using (24), and compute the full charge $Q_3$.

- Impose that $[Q_2, Q_3] = 0$, and solve this equation for the entries of $\mathcal{H}$ and $q$.

- For each solution $\mathcal{H}$, find a Lax operator $\mathcal{L}(u)$ which generates $\mathcal{H}$ via (22) and satisfies the RLL relation (11).

In practice, one needs to embed the operators $Q_2, Q_3$ on a spin chain of large enough length $L$, such that no cancellations occur in $[Q_2, Q_3]$ which do not happen generically. The construction of a Lax operator corresponding to an integrable Hamiltonian is non-trivial. One approach is to make an Ansatz for $\mathcal{L}_{12\ldots r}$

$$\mathcal{L}_{12\ldots r}(u) = \mathcal{P}_{1r}\mathcal{P}_{2r}\cdots\mathcal{P}_{r-1,r}(1 + u\mathcal{H}_{12\ldots r} + O(u^2)) \, , \tag{27}$$

which is motivated since it trivialises the relation (22), and also ensures that the charge $Q_1$ is the momentum generator. From this Ansatz one can form a transfer matrix $t(u)$, and fix the $O(u^2)$ term in (27) by imposing

$$[Q_2, t(u)] = 0 \, . \tag{28}$$

---

[4] In many cases we can choose this operator $q_{12\ldots r} = 0$, i.e. $(\mathcal{H}_{12\ldots r})^2 = \partial_u^2 \check{\mathcal{L}}_{12\ldots r}|_{u=0}$.

The $R$-matrix can then easily be computed by solving the RLL relation (11), since it is a linear equation for $R_{AB}(u,v)$ once $\mathcal{L}$ is known. In the range 2 case, the Lax operator can often be identified with the $R$-matrix. In this case the $R$-matrix can be computed more straightforwardly than solving (28) by solving the *Sutherland equations* [19]. This approach has been used to classify a large class of nearest-neighbour integrable models for local Hilbert space dimensions $d = 2, 3, 4$, and find the corresponding $R$-matrices [20–25]. In the higher-range case there are less results, although an initial step was made in [9] for range 3 and range 4 models.

## 3 Constrained models

In this paper we are studying integrable models on constrained Hilbert spaces. Consider a spin chain with local Hilbert space $V_j$ and length $L$. Suppose the physical space that we want to consider is a subspace of the total Hilbert space $V = \bigotimes_{j=1}^{L} V_j$, and consists of states that satisfy some condition. In particular, we assume that there is some projection operator $\Pi : V \to V$ such that

$$\Pi|\text{physical}\rangle = |\text{physical}\rangle\,, \qquad \Pi|\text{unphysical}\rangle = 0\,. \tag{29}$$

By definition a projection satisfies $\Pi^2 = \Pi$. Then, our physical Hilbert space is given by

$$V_\Pi := \left\{ |v\rangle \in V \,\middle|\, \Pi|v\rangle = |v\rangle \right\}. \tag{30}$$

**Operators.** In order to define operators on our physical Hilbert space we can consider operators which act on the total Hilbert space but that commute with $\Pi$. In other words, we will consider operators $\mathcal{O} : V \to V$ which satisfy

$$[\mathcal{O}, \Pi] = 0\,. \tag{31}$$

This is a natural constraint since this implies that physical and unphysical states do not mix under action of such an operator and the operator can naturally be restricted to $V_\Pi$

$$\mathcal{O}^{(\Pi)} := \Pi\mathcal{O}\Pi = \left(\Pi\mathcal{O} = \mathcal{O}\Pi\right). \tag{32}$$

**Charges.** We define an integrable model on the restricted Hilbert space analogously to the unconstrained case: we require that there is a tower of conserved charges $Q_i$ such that

$$[Q_i, \Pi] = 0\,, \qquad [Q_i^{(\Pi)}, Q_j^{(\Pi)}] = 0 \Leftrightarrow \Pi[Q_i, Q_j] = 0\,. \tag{33}$$

In other words, the charges need only commute on physical states. In section 4 we will describe an equivalent formulation of integrability for these constrained models, based on the construction of a Lax matrix which directly generates the charges $Q_j^{(\Pi)}$ in the specific case of the Rydberg constraint, which we discuss next.

**Rydberg atoms.** In this paper we will focus on the Rydberg constraint. This is the case of a spin-$\frac{1}{2}$ chain where we are not allowed to have two down spins next to each other. This constraint is physically relevant and indeed models of Rydberg atom chains incorporate this constraint, since it is energetically unfavourable to have adjacent excited states [43].

In this case our local Hilbert spaces are $V_j \simeq \mathbb{C}^2 = \text{span}(|\uparrow\rangle, |\downarrow\rangle)$. In order to implement the constraint on the full Hilbert space $V = (\mathbb{C}^2)^{\otimes L}$ we introduce the operators

$$P = \frac{1+Z}{2}\,, \qquad N = \frac{1-Z}{2}\,, \tag{34}$$

which project locally onto up and down spins respectively, where we denote the Pauli matrices by $X, Y, Z$. The two-site local projector which forbids neighbouring down spins is

$$\Pi_{i,i+1} := 1 - N_i N_{i+1}. \tag{35}$$

If we consider periodic boundary conditions, then the projector that encodes the Rydberg constraint on the full chain takes the form

$$\Pi = \prod_{i=1}^{L} \Pi_{i,i+1}. \tag{36}$$

It is easy to check that there are no interesting nearest-neighbour operators that are compatible with this constraint. This is the reason we consider operators of range $r \geq 3$. The dimension of the constrained Hilbert space grows as powers of the golden ratio

$$\dim V_\Pi \sim \phi^L. \tag{37}$$

For example, for $L = 4$ we have

$$V_\Pi = \mathrm{span}(|\uparrow\uparrow\uparrow\uparrow\rangle, |\downarrow\uparrow\uparrow\uparrow\rangle, |\uparrow\downarrow\uparrow\uparrow\rangle, |\uparrow\uparrow\downarrow\uparrow\rangle, |\uparrow\uparrow\uparrow\downarrow\rangle, |\uparrow\downarrow\uparrow\downarrow\rangle, |\downarrow\uparrow\downarrow\uparrow\rangle). \tag{38}$$

**Range and gauge symmetry.** Constraints such as (36) destroy the product form of the Hilbert space $V$. Because of this, it is subtle to define the notion of range of an operator. On the full Hilbert space $V$ the range of a translationally invariant operator $\mathcal{O}$ is usually formulated at the level of operator densities. In particular, the range of $\mathcal{O}$ is the minimal $r$ such that

$$\mathcal{O} = \sum_{i=1}^{L} \mathcal{O}_{i,i+1,\dots,i+r-1}, \tag{39}$$

for some operator density $\mathcal{O}_{i,i+1,\dots,i+r-1}$. We take the minimal $r$ because $\mathcal{O}_{i,i+1,\dots,i+r-1}$ can always be trivially extended by adjoining identity operators.

On the constrained Hilbert space $V_\Pi$ this is a bit more subtle. Here there can be operator densities of apparently different range, which lead to the same total operator on the constrained Hilbert space $V_\Pi$. For example, we have that

$$\Pi\left[\sum_{i=1}^{L} P_i N_{i+1} P_{i+2}\right]\Pi = \Pi\left[\sum_{i=1}^{L} -P_i P_{i+1} + P_i\right]\Pi, \tag{40}$$

so that this operator can apparently be equivalently be represented in terms of a range 3 or a range 2 operator density. Equation (40) is just the statement that on the Rydberg-blockaded chain one can count the number of excitations $|\uparrow\downarrow\uparrow\rangle$ directly with the operator $PNP$, or indirectly by counting the total number of up spins $|\uparrow\rangle$, and subtracting the number of adjacent pairs $|\uparrow\uparrow\rangle$. A similar identity relates a range 4 operator density to a range 3 one:

$$\Pi\left[\sum_{i=1}^{L} \frac{1}{2}\left(P_i P_{i+1} N_{i+2} P_{i+3} + P_i N_{i+1} P_{i+2} P_{i+3}\right)\right]\Pi = \Pi\left[\sum_{i=1}^{L} -P_i P_{i+1} P_{i+2} + P_i P_{i+1}\right]\Pi. \tag{41}$$

Identities such as (40) and (41) lead to a large gauge freedom in defining Hamiltonian density operators on the constrained space. Such identities can be non-trivial and must be taken into account when counting the number of independent operators at each range.

For later sections we will require a consistent definition of range for operators on the constrained Hilbert space $V_\Pi$. It can be shown that any operator on $V_\Pi$ can be written as a sum of operators of the form

$$\mathcal{O}_r^{(\Pi)} := \Pi\Big[\sum_{i=1}^{L} P_i \mathcal{O}_{i+1,i+2,\dots,i+r-2} P_{i+r-1}\Big]\Pi\,, \tag{42}$$

where $\mathcal{O}_{i+1,i+2,\dots,i+r-2}$ is a usual operator density of range $r-2$. This can be seen by noting that any $2 \times 2$ matrix $A$ can be decomposed as

$$A = a_x X + a_y Y + a_p P + a_n N\,. \tag{43}$$

Let's now look at the rightmost factor in $\mathcal{O}$ when decomposed as a sum of tensor products of $2 \times 2$ matrices. We can assume that $a_p = 0$, since if it is non-zero, then that part of the operator is already of the right form. We then have the following identity

$$\Pi_{i,i+1} A_i \Pi_{i,i+1} = \Pi_{i,i+1} A_i P_{i+1} \Pi_{i,i+1}\,, \tag{44}$$

hence we can always assume that the right-most factor of the constrained operator is $P$. Similar arguments hold for the leftmost term. We will define the range of a constrained operator $\mathcal{O}^{(\Pi)}$ to be the minimal $r$ such that $\mathcal{O}^{(\Pi)}$ admits a representation as

$$\mathcal{O}^{(\Pi)} = \sum_{r'=1}^{r} \mathcal{O}_{r'}^{(\Pi)}\,. \tag{45}$$

We stress that the form (45) of an operator on $V_\Pi$ may not be the simplest available. However, for consistency we take this as our initial Ansatz for range $r$ models in later sections. After we classify the integrable models for each range $r$, we can use identities such as (40), (41), and $P + N = 1$ to search for a potentially more natural form of the Hamiltonians.

## 4 Constrained integrability

In this section we outline how to adapt the higher-range integrability formulation discussed in section 2 to include models with the Rydberg constraint. We first discuss the RLL formulation of integrability in this constrained setting. Then, we discuss a modified formulation of integrability based on the so-called GLL relation. This formulation makes manifest the connection between Rydberg constrained models and IRF models in statistical mechanics, which we discuss in more detail in section 5. Finally, we will outline our approach for classifying integrable Rydberg-constrained models by range, according to the definition (45). In this section we focus on generalities, and give details for specific integrable constrained models in later sections.

### 4.1 Constrained RLL

**Lax operator.** Due to (45), a Rydberg-constrained model of range $r$ can be written as

$$Q_2^{(\Pi)} = \Pi\left[\sum_{i=1}^{L} \mathcal{H}_{i,i+1,\dots,i+r-1}\right]\Pi\,, \tag{46}$$

where $\mathcal{H}_{i,i+1,\dots,i+r-1}$ is a Hamiltonian density of range $r$ in the usual sense.[5] A possible way to define the integrability of the model (46) would be to apply the results of section 2 for the

---

[5]In general $\mathcal{H}_{i,i+1,\dots,i+r-1}$ can contain lower range parts. There is also a large gauge freedom in defining the density $\mathcal{H}$, due to periodicity and redundancies in the constrained Hilbert space.

Hamiltonian density $\mathcal{H}_{12...r}$, modifying them slightly to account for the Rydberg constraint. In particular, we can require the existence of a Lax operator

$$\mathcal{L}_{12...r}(u) = \mathcal{P}_{1r}\mathcal{P}_{2r}\cdots\mathcal{P}_{r-1,r}(1 + u\mathcal{H}_{12...r} + O(u^2)), \tag{47}$$

such that the charges generated from the corresponding transfer matrix commute on the constrained subspace

$$[Q_i^{(\Pi)}, Q_j^{(\Pi)}] = 0. \tag{48}$$

The charges (48) are derived via logarithmic derivatives of the transfer matrix on the constrained subspace

$$Q_j^{(\Pi)} = \frac{d^{j-1}}{du^{j-1}} \log t^{(\Pi)}(u)\Big|_{u=0}, \tag{49}$$

where

$$t^{(\Pi)}(u) = \Pi \, \text{tr}_A[\mathcal{L}_{AL}(u)\mathcal{L}_{A,L-1}(u)\cdots\mathcal{L}_{A1}(u)] \, \Pi. \tag{50}$$

In (50) we use the multi-index $A = (a_1 a_2 \cdots a_{r-1})$, which reflects the auxiliary space decomposition $V_A \simeq V_{a_1} \otimes V_{a_2} \otimes \cdots \otimes V_{a_{r-1}}$. The precise form of (47) ensures that the first charge $Q_1^{(\Pi)}$ is the momentum operator and the second charge $Q_2^{(\Pi)}$ coincides with (46). The higher charge $Q_3^{(\Pi)}$ can be constructed directly from the Hamiltonian density by applying a projection to (24):

$$Q_3^{(\Pi)} = \Pi\left[\sum_{i=1}^{L}\left([\mathcal{H}_{i,i+1,...,i+r-1}, \mathcal{H}_{i+1,i+2...,i+r} + \ldots + \mathcal{H}_{i+r,i+r+1,...,i+2r-1}] + q_{i,i+1,...,i+r-1}\right)\right]\Pi, \tag{51}$$

where $q_{i,i+1,...,i+r-1}$ is some operator density of range $r$. Given a Lax operator (47), one would hope to prove the integrability condition (48) by establishing the commutation of transfer matrices at different values of the spectral parameter

$$[t^{(\Pi)}(u), t^{(\Pi)}(v)] = 0, \tag{52}$$

which itself would follow from an RLL relation

$$R_{AB}(u,v)\mathcal{L}_{Ai}(u)\mathcal{L}_{Bi}(v) = \mathcal{L}_{Bi}(v)\mathcal{L}_{Ai}(u)R_{AB}(u,v), \tag{53}$$

for some $R$-matrix $R_{AB}(u,v)$. At this stage there is an issue, however, since $\mathcal{L}$ contains no information about the Rydberg constraint we are considering. Therefore any solution to (53) would necessarily also correspond to an unconstrained integrable model, i.e. the Lax matrix would generate an infinite tower of charges $Q_i$ which mutually commute on the whole Hilbert space $V = (\mathbb{C}^2)^{\otimes L}$. In general, an integrable Hamiltonian $Q_2$ on $V$ will not commute with the projection $\Pi$, and so cannot be consistently restricted to an integrable model on $V_\Pi$. Therefore, given a model which is only integrable on the constrained subspace $V_\Pi$, there can be no solution to (53), and we need a modified formulation of integrability in order to prove (52).

**Projected Lax operator.** We circumvent this issue by constructing a Lax matrix which directly generates the transfer matrix on the constrained subspace. Using this modified Lax matrix we will be able to propose a corresponding RLL relation, from which the relation (52) can be proven. This is possible if we define

$$\tilde{\mathcal{L}}_{Aj}(u) := \Pi_A \mathcal{L}_{Aj}(u), \tag{54}$$

where $\mathcal{L}_{Aj}(u)$ is the Lax operator (47). In (54) we inserted an open projector

$$\Pi_A := \Pi_{a_1 a_2}\Pi_{a_2 a_3}\cdots\Pi_{a_{r-2}a_{r-1}}, \tag{55}$$

on the auxiliary space indices. This projected Lax matrix directly generates the transfer matrix acting on the constrained subspace

$$\tilde{t}(u) := \text{tr}_A[\tilde{\mathcal{L}}_{AL}(u)\tilde{\mathcal{L}}_{A,L-1}(u)\cdots\tilde{\mathcal{L}}_{A1}(u)] = t^{(\Pi)}(u). \tag{56}$$

This result can be proven as follows. First of all, by (45) the Hamiltonian density $\mathcal{H}$ can be written as a sum of operators of the form $P\mathcal{O}P$, and for compatibility with the Rydberg constraint we further require

$$\left[\sum_{i=1}^{L}\mathcal{H}_{i,i+1,\dots,i+r-1},\Pi\right] = 0. \tag{57}$$

Explicit computation shows that this implies that the Hamiltonian density itself commutes with the open projector with one site removed

$$[\mathcal{H}_{12\cdots r},\Pi_{12}\Pi_{23}\cdots\Pi_{r-2,r-1}] = 0. \tag{58}$$

The same is true for the higher order terms in the Lax operator (47), so the commutation relations of $\Pi_A$ and $\mathcal{L}_{Aj}$ are entirely determined by the permutation operators explicitly included in (47).

For example, consider the case $r = L = 3$. The projected Lax matrix in this case is

$$\tilde{\mathcal{L}}_{a_1a_2j}(u) = \Pi_{a_1a_2}\mathcal{L}_{a_1a_2j}(u) = \Pi_{a_1a_2}\mathcal{P}_{a_1j}\mathcal{P}_{a_2j}(1 + u\mathcal{H}_{a_1a_2j} + O(u^2)), \tag{59}$$

where the sum of operators in the brackets commutes with the projector $\Pi_{a_1a_2}$. Then we have

$$\tilde{t}(u) = \text{tr}_{a_1a_2}[\Pi_{a_1a_2}\mathcal{L}_{a_1a_23}(u)\Pi_{a_1a_2}\mathcal{L}_{a_1a_22}(u)\Pi_{a_1a_2}\mathcal{L}_{a_1a_21}(u)]. \tag{60}$$

Since the commutation relations of $\Pi_{a_1a_2}$ and $\mathcal{L}_{a_1a_2j}(u)$ are the same as those of $\Pi_{a_1a_2}$ and $\mathcal{P}_{a_1j}\mathcal{P}_{a_2j}$, we can commute the projectors to the right

$$\begin{aligned}
\tilde{t}(u) &= \text{tr}_{a_1a_2}[\mathcal{L}_{a_1a_23}(u)\mathcal{L}_{a_1a_22}(u)\mathcal{L}_{a_1a_21}(u)\Pi_{a_21}]\Pi_{12}\Pi_{23} \\
&= \text{tr}_{a_1a_2}[\mathcal{L}_{a_1a_23}(u)\mathcal{L}_{a_1a_22}(u)\mathcal{L}_{a_1a_21}(u)]\Pi_{12}\Pi_{23}\Pi_{31} \\
&= t(u)\Pi = t^{(\Pi)}(u). 
\end{aligned} \tag{61}$$

The second equality in (61) can be easily seen by replacing all the Lax operators with the corresponding permutation operators $\mathcal{L}_{a_1a_2j} \sim \mathcal{P}_{a_1j}\mathcal{P}_{a_2j}$, and the fourth equality uses the fact that an operator $\mathcal{O}$ compatible with the Rydberg constraint satisfies $\Pi\mathcal{O}\Pi = \mathcal{O}\Pi = \Pi\mathcal{O}$. The proof of (56) proceeds analogously for higher $r$ and $L$.

**Projective integrability.** With a projected Lax operator $\tilde{\mathcal{L}}$ which directly generates the projected transfer matrix $t^{(\Pi)}(u)$, we can proceed to adapt the integrability machinery of section 2 to the case of the Rydberg constraint. In order to prove the integrability condition (52), we look for an $R$-matrix $R_{AB}(u,v)$ such that a projected analogue of the RLL relation is satisfied

$$R_{AB}(u,v)\tilde{\mathcal{L}}_{Ai}(u)\tilde{\mathcal{L}}_{Bi}(v) = \tilde{\mathcal{L}}_{Bi}(v)\tilde{\mathcal{L}}_{Ai}(u)R_{AB}(u,v). \tag{62}$$

We note that due to the projectors in $\tilde{\mathcal{L}}$, many of the entries of $R_{AB}$ will be projected away and therefore not fixed by (62). We can therefore require that $R_{AB}$ acts on an appropriate projected subspace of $V_A \otimes V_B$ by defining

$$\tilde{R}_{AB}(u,v) = \Pi_{AB}R_{AB}(u,v), \tag{63}$$

where $\Pi_{AB}$ is an appropriate projector. We require this projector to be such that the corresponding Yang–Baxter equation is satisfied

$$\tilde{R}_{AB}(u,v)\tilde{R}_{AC}(u,w)\tilde{R}_{BC}(v,w) = \tilde{R}_{BC}(v,w)\tilde{R}_{AC}(u,w)\tilde{R}_{AB}(u,v), \tag{64}$$

as well as a projected version of braiding unitarity

$$\tilde{R}_{AB}(u,v)\tilde{R}_{BA}(v,u) \propto \Pi_{AB}. \tag{65}$$

Finally, we require that the projector $\Pi_{AB}$ is compatible with the projected monodromy matrix

$$\tilde{\Pi}_{BA}\tilde{T}_A(u)\tilde{T}_B(v) = \tilde{T}_A(u)\tilde{T}_B(v), \qquad \tilde{T}_B(v)\tilde{T}_A(u)\tilde{\Pi}_{AB} = \tilde{T}_B(v)\tilde{T}_A(u), \tag{66}$$

where

$$\tilde{T}_A(u) := \tilde{\mathcal{L}}_{AL}(u)\tilde{\mathcal{L}}_{A,L-1}(u)\cdots\tilde{\mathcal{L}}_{A1}(u). \tag{67}$$

We note that (66) is not a strong condition since there are already explicit projectors in $\tilde{T}_A$ and $\tilde{T}_B$. We will give explicit expressions for $\Pi_{AB}$ for range 3 and range 4 models in the coming sections. By combining these properties we will be able to prove the integrability condition (52). From the RLL relation (62) it is straightforward to derive a projected analogue of the RTT relation (16)

$$\tilde{R}_{AB}(u,v)\tilde{T}_A(u)\tilde{T}_B(v) = \tilde{T}_B(v)\tilde{T}_A(u)\tilde{R}_{AB}(u,v). \tag{68}$$

Since the projected $R$-matrix $\tilde{R}$ is not invertible, we cannot prove $[t^{(\Pi)}(u), t^{(\Pi)}(v)] = 0$ from (68) in the usual way. However, we can multiply both sides of this equation on the left by $\tilde{R}_{BA}(v,u)$, and use braiding unitarity (65) and cyclicity of the trace to derive

$$\text{tr}_{AB}[\tilde{\Pi}_{BA}\tilde{T}_A(u)\tilde{T}_B(v)] = \text{tr}_{AB}[\tilde{T}_B(v)\tilde{T}_A(u)\tilde{\Pi}_{AB}]. \tag{69}$$

Finally, we can use (66) to conclude

$$[t^{(\Pi)}(u), t^{(\Pi)}(v)] = 0, \tag{70}$$

proving the integrability of the model.

## 4.2  Constrained GLL

There is an alternative formulation of integrability for higher-range integrable models, in the special case where the checked Lax operator $\check{\mathcal{L}}(u)$ takes the form

$$\check{\mathcal{L}}(u) := \mathcal{P}_{r-1,r}\cdots\mathcal{P}_{1r}\mathcal{L}_{12\ldots r}(u) = \sum_{j,k=\circ,\bullet} P_1^j \mathcal{O}_{23\cdots r-1}^{jk}(u) P_r^k, \tag{71}$$

where $P^\circ = P$ and $P^\bullet = N$. In other words, the first and last operators in the checked Lax are control bits. In this formulation the $R$-matrix is replaced by the $G$-operator, which contains the same information as $R$ but acts on one less site. This formulation exposes the relation between these types of integrable models and IRF models in statistical mechanics. Since we can write the Hamiltonian of a constrained integrable model as a sum of operators of the form $P\mathcal{O}P$, the corresponding Lax matrix admits the decomposition (71) in a trivial way. Therefore this formalism can be adapted to integrable constrained models. We describe the derivation of the $G$ operator for range 3 and 4 integrable constrained models.

**Range 3.** We begin with the projected RLL relation

$$R_{AB}(u,v)\tilde{\mathcal{L}}_{Ai}(u)\tilde{\mathcal{L}}_{Bi}(v) = \tilde{\mathcal{L}}_{Bi}(v)\tilde{\mathcal{L}}_{Ai}(u)R_{AB}(u,v), \tag{72}$$

for range 3 models. In this case the auxiliary space contains two copies of $\mathbb{C}^2$, so we write the multi-indices $A = (a_1 a_2)$ and $B = (b_1 b_2)$. We write everything in 'checked' form, i.e. we factor out permutation operators from $R_{AB} = \mathcal{P}_{AB}\check{R}_{AB}$ and $\tilde{\mathcal{L}}_{a_1 a_2 j} = \Pi_{a_1 a_2}\mathcal{L}_{a_1 a_2 j} = \Pi_{a_1 a_2}\mathcal{P}_{a_1 j}\mathcal{P}_{a_2 j}\check{\mathcal{L}}_{a_1 a_2 j}$. Due to the $\mathcal{POP}$ structure, the Lax operator satisfies the relation

$$[\check{\mathcal{L}}_{123}(u), \check{\mathcal{L}}_{345}(v)] = 0. \tag{73}$$

Ordering the spaces $(jAB) = (ja_1 a_2 b_1 b_2) = (12345)$ equation (72) can be rewritten

$$\check{\mathcal{L}}_{345}(u)^{-1}\check{R}_{2345}(u,v)\check{\mathcal{L}}_{345}(v)\Pi_{23}\Pi_{45} = \Pi_{23}\Pi_{45}\check{\mathcal{L}}_{123}(v)\check{R}_{1234}(u,v)\check{\mathcal{L}}_{123}(u)^{-1}. \tag{74}$$

Since the left hand side of (74) acts trivially on the first space, and the right hand side acts trivially on the fifth space, it implies that both sides of the equation are equal to

$$\Pi_{23}\Pi_{45}\check{G}_{234}(u,v)\Pi_{23}\Pi_{45}, \tag{75}$$

for some range 3 operator $\check{G}_{234}(u,v)$. This implies two ways of writing the $R$ matrix in terms of the operator $\check{G}$, namely

$$\check{R}_{2345}(u,v)\Pi_{23}\Pi_{45} = \Pi_{23}\Pi_{45}\check{\mathcal{L}}_{345}(u)\check{G}_{234}(u,v)\check{\mathcal{L}}_{345}^{-1}(v)\Pi_{23}\Pi_{45}, \tag{76}$$

$$\Pi_{23}\Pi_{45}\check{R}_{1234}(u,v) = \Pi_{23}\Pi_{45}\check{\mathcal{L}}_{123}(v)^{-1}\check{G}_{234}(u,v)\check{\mathcal{L}}_{123}(u)\Pi_{23}\Pi_{45}. \tag{77}$$

Relabelling the indices on the first of these equations and acting with appropriate projectors, we conclude the following GLL relation

$$\check{G}_{234}(u,v)\check{\mathcal{L}}_{123}(u)\check{\mathcal{L}}_{234}(v)\Pi_{\text{open}} = \Pi_{\text{open}}\check{\mathcal{L}}_{123}(v)\check{\mathcal{L}}_{234}(u)\check{G}_{123}(u,v), \tag{78}$$

where $\Pi_{\text{open}} := \Pi_{12}\Pi_{23}\Pi_{34}\Pi_{45}$. The projectors in (78) imply that there are several undetermined entries in $\check{G}$. We can consistently set them to zero by defining

$$\tilde{G}_{123} = \Pi_{12}\Pi_{23}\check{G}_{123}. \tag{79}$$

In this case the $G$ operator satisfies a regularity condition

$$\tilde{G}_{123}(u,u) = \Pi_{12}\Pi_{23}, \tag{80}$$

and is related to the Lax operator via

$$\tilde{G}_{123}(0,v) = \Pi_{12}\Pi_{23}\check{\mathcal{L}}_{123}(v)^{-1}, \tag{81}$$

$$\tilde{G}_{123}(u,0) = \Pi_{12}\Pi_{23}\check{\mathcal{L}}_{123}(u). \tag{82}$$

The $G$ operator further satisfies a braiding unitarity relation

$$\tilde{G}_{123}(u,v)\tilde{G}_{123}(v,u) = \Pi_{12}\Pi_{23}, \tag{83}$$

and the Yang–Baxter equation

$$\tilde{G}_{234}(u,v)\tilde{G}_{123}(u,w)\tilde{G}_{234}(v,w) = \tilde{G}_{123}(v,w)\tilde{G}_{234}(u,w)\tilde{G}_{123}(u,v). \tag{84}$$

The $G$ operator contains all the information of the $R$ matrix for these range 3 models of the $\mathcal{POP}$ type, but acts on one less site. Analogously to the RLL relation, the GLL relation (78) can be used to prove the commutation of transfer matrices $[t^{(\Pi)}(u), t^{(\Pi)}(v)] = 0$.

**Range 4.** The derivation of the $G$ operator for range 4 models proceeds analogously to above, so we describe it briefly. Here the auxiliary space contains three copies of $\mathbb{C}^2$, and we order the spaces as $(jAB) = (ja_1 a_2 a_3 b_1 b_2 b_3) = (1234567)$. In this case the Lax operator satisfies the relation

$$\left[ \check{\mathcal{L}}_{1234}(u), \check{\mathcal{L}}_{4567}(v) \right] = 0. \tag{85}$$

Similarly to the range 3 case, we rewrite the RLL relation (72) in checked form, and use (85) to conclude the existence of a 5 site operator $\check{G}_{12345}(u)$ which satisfies

$$\check{G}_{23456}(u,v)\check{\mathcal{L}}_{1234}(u)\check{\mathcal{L}}_{3456}(v)\Pi_{\text{open}} = \Pi_{\text{open}}\check{\mathcal{L}}_{1234}(v)\check{\mathcal{L}}_{3456}(u)\check{G}_{12345}(u,v), \tag{86}$$

where in this case $\Pi_{\text{open}} = \Pi_{12}\Pi_{23}\cdots\Pi_{67}$. This 5-site $G$ operator contains the same information as the $R$-matrix, and can be used to prove the integrability condition $[t^{(\Pi)}(u), t^{(\Pi)}(v)] = 0$.

### 4.3 Classification of integrable Rydberg-constrained models

The above results allow for a method to classify integrable Rydberg-constrained models by range. Since the Hilbert space is not of product form we need a consistent notion of range in this case: we use the definition (45). A general range $r$ model on $V_\Pi$ can be written as

$$Q_2^{(\Pi)} = \Pi\left( \sum_{i=1}^{L} \mathcal{H}_{i,i+1,\ldots,i+r-1} \right)\Pi, \tag{87}$$

where $\mathcal{H}$ can be written as a sum of operators of the form $P\mathcal{O}P$:

$$\mathcal{H}_{i,i+1,\ldots,i+r-1} = \sum_{r'=1}^{r} P_i \mathcal{O}_{i+1,\ldots,i+r'-2} P_{i+r'-1}. \tag{88}$$

For $r' = 1,2$ we take the operators to be $P_i$ and $P_i P_{i+1}$ respectively. This model is integrable if there exists a Lax matrix $\mathcal{L}_{Aj}(u)$ which generates $Q_2^{(\Pi)}$ and an infinite number of mutually commuting charges $Q_j^{(\Pi)}$. In particular we have

$$\left[ Q_2^{(\Pi)}, Q_3^{(\Pi)} \right] = 0, \tag{89}$$

where $Q_3^{(\Pi)}$ can be calculated from the Hamiltonian density using (51). The condition (89) is a necessary condition for integrability. However, as discussed in section 2, it appears to be sufficient. We solve (89) to find the integrable Rydberg-constrained Hamiltonians for each range $r$. To summarise, our method is as follows:

- Parametrise a general Rydberg-constrained Hamiltonian density $\mathcal{H}_{i,i+1,\ldots,i+r-1}$ using (88), potentially requiring for some extra symmetries such as time- or space-reflection invariance.

- Compute the total Hamiltonian $Q_2^{(\Pi)} = \Pi\left( \sum_{i=1}^{L} \mathcal{H}_{i,i+1,\ldots,i+r-1} \right)\Pi$.

- Compute the higher charge $Q_3^{(\Pi)}$ from (51) in terms of $\mathcal{H}$ and an undetermined range $r$ operator $q_{12\ldots r}$.

- Impose $[Q_2^{(\Pi)}, Q_3^{(\Pi)}] = 0$, and solve the resulting set of equations for $\mathcal{H}$ and $q$.

- For each solution, verify integrability by constructing a projected Lax operator $\tilde{\mathcal{L}}_{Aj}$ which generates $Q_2^{(\Pi)}$ and satisfies an appropriate $\tilde{R}\tilde{L}\tilde{L}$ relation.

As before, the operators $Q_2^{(\Pi)}$ and $Q_3^{(\Pi)}$ need to be embedded on a chain of large enough length $L$, such that no cancellations occur in $[Q_2^{(\Pi)}, Q_3^{(\Pi)}]$ which do not happen generically. We apply this procedure for Rydberg-constrained models of range 3, 4, and 5 in sections 6, 7, and appendix A.

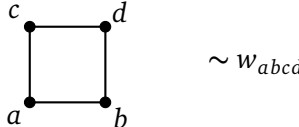

Figure 1: Depiction of a plaquette. The variables $a, b, c, d$ can take values 0 and 1. We have the Rydberg constraints for the pairs $ab$, $ac$, $bd$ and $cd$. The weight of the plaquette is denoted as $w_{abcd}$.

# 5   Interpretation via 2D statistical physics models

Here we provide a correspondence between 1D quantum Hamiltonians acting on constrained Hilbert spaces, and 2D statistical physics models on square lattices, where the same type of constraints are applied for the classical configurations. Famous examples for such classsical models are the Restricted Solid-on-Solid (RSOS) models [45, 46]. We extend known connections [44] to models with medium-range interactions. In all examples below we treat only the Rydberg constraint, but the extension to other types of constraints is straightforward.

In the 2D classical models we consider a square lattice of size $L \times M$, where $L$ is the number of sites in the "horizontal" direction. This will correspond to the space direction, and $L$ is equal to the length of the associated quantum spin chain. The vertical direction will be seen as the imaginary time direction. In principle we can consider periodic and open boundary conditions too, but in our examples we will focus on the periodic cases.

In the classical models there is a dynamical variable associated with every vertex of the square lattice. In our examples these variables can take values 0 and 1. The Rydberg constraint means that we allow those configurations on the lattice which don't have two 1's at neighbouring positions in the lattice. This constraint holds for the nearest-neighbours in both the horizontal and vertical directions. However, there is no constraint for the diagonal directions. For models with constraints also in the diagonal directions see [78, 79].

First we review the known RSOS construction, which leads to the constrained quantum spin chains with 3-site interactions; our treatment is similar to that of [44]. Afterward we extend the ideas to the 4-site interacting case.

## 5.1   Range 3

In the model the partition function is given by the sum of the Boltzmann weights of the allowed configurations

$$Z = \sum_{\text{allowed}} w_{\{s\}}, \tag{90}$$

and the individual weights are given by a product over weights associated to plaquettes:

$$w_{\{s\}} = \prod_{\text{plaquettes}} w_P, \tag{91}$$

where every $w_P$ is a weight associated to a certain elementary plaquette. These weights depend on the 4 variables on the vertices. Denoting them as $a, b, c, d$ (see fig. 1) we can write

$$w_P = w_{abcd}(u), \tag{92}$$

where $u$ is the spectral paramater.

The weights satisfy a certain type of Yang-Baxter relation. First let us introduce a new object $G(u, v)$, which also describes plaquette weights. It also depends on four local variables,

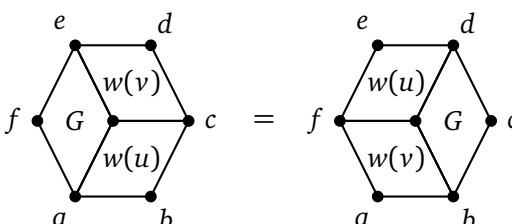

Figure 2: Yang-Baxter relation for the plaquette weights. It is understood that there is a summation for the middle vertex, while keeping the boundary vertices fixed. The weights $G$ depend on $u$ and $v$, but this is not denoted explicitly.

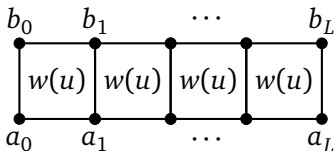

Figure 3: Monodromy matrix for total length $L + 1$, acting from the lower set of variables to the upper set. The value of a certain component is the product of the plaquette weights $w(u)$. The periodic transfer matrix is obtained by allowing configurations only with $a_0 = a_L$ and $b_0 = b_L$.

and the components are denoted as $G_{abcd}(u, v)$. In certain cases these weights will be identical to those given by $w(u - v)$, but this is not necessary. This object essentially coincides with the operator $\check{G}$ introduced in Section 4.2, but here we introduce it independently, following the logic of the statistical physics models at hand.

The Yang-Baxter relation in question is given pictorially by Fig. 2. Explicitly it reads

$$\sum_s G_{asef}(u, v) w_{abcs}(u) w_{scde}(v) = \sum_s w_{absf}(v) w_{fsde}(u) G_{bcds}(u - v). \tag{93}$$

We build a "monodromy matrix" and a "transfer matrix". They both come from a concatenation of plaquettes in the horizontal direction. We use the same spectral parameter along the lattice, although this is not necessary.

We denote the monodromy matrix by $T(u)$. It is a matrix acting on the restricted Hilbert space with $L + 1$ sites *with free boundary conditions*. The concrete matrix elements are given by

$$T^{b_0 b_1 \cdots b_L}_{a_0 a_1 \cdots a_L}(u) = \prod_{j=0}^{L-1} w_{a_j a_{j+1}, b_j b_{j+1}}(u). \tag{94}$$

A pictorial depiction of the monodromy matrix for total length $L + 1$ is given in fig 3.

In contrast, we denote the transfer matrix by $t(u)$, and this is a matrix that acts on the restriced Hilbert space with $L$ sites. The difference with respect to the monodromy matrix is that now we impose the periodicity $a_0 = a_L$ and $b_0 = b_L$, but keeping the formal structure of the weights the same. The concrete formula is

$$t^{b_1 \cdots b_L}_{a_1 \cdots a_L}(u) = \prod_{j=0}^{L-1} w_{a_j a_{j+1}, b_j b_{j+1}}(u), \tag{95}$$

where it is now understood that $a_0 = a_L$ and $b_0 = b_L$. Note that the step of going from $T(u)$ to $t(u)$ is not the same as taking a trace over an auxiliary space: there is no additional summation, and the concrete values of the matrix elements of $t(u)$ are actually equal to the values of the



Figure 4: The definition of the inverse of $G$. The dashed line signals that those values are identical.

selected matrix elements of $T(u)$. The only difference between the two matrices is that $T(u)$ also acts on configurations which are not allowed for $t(u)$.

The partition function (90) of the periodic lattice with horizontal size $L$ and vertical size $M$ is then calculated as

$$Z = \text{Tr}(t(u))^M \,. \tag{96}$$

Here it is understood that the product of transfer matrices is evaluated in the usual way:

$$\left(t(v)t(u)\right)_{a_1 \cdots a_L}^{c_1 \cdots c_L} = \sum_{b_1 \cdots b_L} t_{b_1 \cdots b_L}^{c_1 \cdots c_L}(v) t_{a_1 \cdots a_L}^{b_1 \cdots b_L}(u) \,. \tag{97}$$

A certain type of "train argument" can be used to prove the commutativity

$$[t(u), t(v)] = 0 \,. \tag{98}$$

The train argument can be applied with the insertion of an additional plaquette, described by the weights $G(u, v)$, together with an inversion relation for $G$. In simpified notation the inversion relation is $G^{-1} \cdot G = G \cdot G^{-1} = 1$, see Fig. 4. In more concrete terms the inversion is defined formally as

$$\sum_s G_{asbc}(u) G_{adbs}^{-1}(u) = \sum_s G_{asbc}^{-1}(u) G_{adbs}(u) = \delta_{cd} \,. \tag{99}$$

In certain cases cases we have $G_{abcd}^{-1}(u) = G_{abcd}(-u)$, but this is not required.

We insert this inversion relation into the product $t(u)t(v)$. We can commute for example $g$ through the two rows, using the Yang-Baxter relation (93). Once we end up with coming back from "the other side", we use the inversion relation again and finally obtain $t(v)t(u)$.

Now we also construct the quantum spin chains associated to these models. We derive local Hamiltonians which commute with the family $t(u)$. As usual, the Hamiltonian is obtained as a first logarithmic derivative of $t(u)$.

First, we interpret the elementary plaquettes with weights $w_{abcd}$ as a linear operator acting on 3 sites. This linear operator will be such that it acts diagonally on the first and the third sites, and has a non-trivial action in the middle site. The identification is found by considering an action from the SW corner to the NE corner of the plaquette in Fig. 1. This gives

$$w_{abcd} \quad \rightarrow \quad \check{\mathcal{L}}(u) = \sum_{a,b,c,d} w_{abcd}(u) \times \left( P_c \otimes (|d\rangle\langle a|) \otimes P_b \right) \,. \tag{100}$$

Now $\check{\mathcal{L}}$ is a three site operator, $P_b$ and $P_c$ are local projectors to the basis states. The operator in the middle has typically off-diagonal elements as well. It is understood that the summation is over the allowed plaquette configurations only. The operator $\check{\mathcal{L}}$ coincides with the Lax operator introduced in earlier sections, but now we introduced it from the statistical physics side. Afterwards we perform a similar identification for the weights given by $G$, to arrive at the linear operators $\check{G}$ introduced earlier.

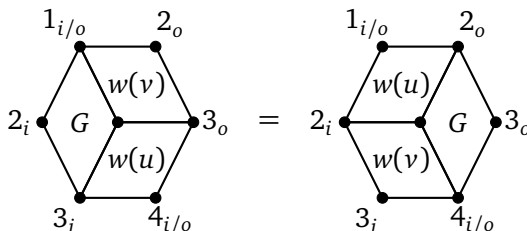

Figure 5: Yang-Baxter relation with quantum spaces identified.

Let us now re-interpret the Yang-Baxter relation (93) as an equality for the linear operators. The relation will be found as an equality of a certain operator product acting on 4 quantum spaces. In order to find a direct correspondence we redraw the pictorial interpretation of (93), and we indentify "incoming" and "outgoing" vector spaces. The role of the vector space will be denoted with subscripts. The resulting operator product will act diagonally on the first and last space, therefore in those cases we write $1_{i/o}$ and $4_{i/o}$. In this way we obtain Fig. 5. We can now read off the linear equation, and we get

$$\check{\mathcal{L}}_{123}(v)\check{\mathcal{L}}_{234}(u)\check{G}_{123}(u,v) = \check{G}_{234}(u,v)\check{\mathcal{L}}_{123}(u)\check{\mathcal{L}}_{234}(v)\,. \tag{101}$$

This is essentially the same equation as (78), with the only exception that here we did not introduce the projectors enforcing the constraint. However, the constraint was included implicitly, because we are considering only the allowed configurations. (101) formally coincides with (V.10) from [9], but that work considered similar systems without constraints.

The spin chain Hamiltonians are obtained in the usual way. The regularity condition means that $\check{\mathcal{L}}(u=0)$ is the identity operator. This means that at $u=0$ the only allowed plaquettes are such that $a=d$ for all $b,c$, and they come with weight 1. Then $t(u)$ becomes the shift operator. Taking the first logarithmic derivative we get the Hamiltonians, with the usual formula

$$H = \sum_j h_j\,, \qquad h_j = \partial_u \check{\mathcal{L}}_{j,j+1,j+2}(u)|_{u=0}\,. \tag{102}$$

## 5.2 Range 4

Now we construct 2D statistical physics models which correspond to the Hamiltonians with range 4 interactions. To our best knowledge interaction-round-a-face models with medium-range interactions have not yet been published in the literature, but a somewhat related model is treated in [59].

We tile the square lattice with rectangles of size $1 \times 2$, and we attach statistical weights to each rectangle. These weights depend on the 6 variables along the sides of the rectangle, see Fig 6. We construct the monodromy matrix and the transfer matrix via the horizontal concatenation of the rectangles. Once again, the distinction between the two matrices is only regarding the boundary conditions. For the transfer matrix see Fig. 7.

Once again we introduce a certain auxiliary object $G$, which will be used to prove the commutativity of the transfer matrices. We find that the "size" of $G$ differs from that of the elementary rectangles. In this case $G$ describes the weight of a square of size $2 \times 2$, so that the weight depends on the values of 8 variables along the sides of the square. For a graphical interpretation see Fig. 8, where we draw the square in a slightly deformed way. This way of depicting $G$ is convenient for the purpose of formulating the Yang-Baxter relation, which is a relation that intertwines two elementary rectangles with weights $w(u)$ and $w(v)$. The relation is depicted on Fig. 9. This relation can be used to prove the commutativity of the transfer matrices in the medium-range case, via a direct application of the train argument.

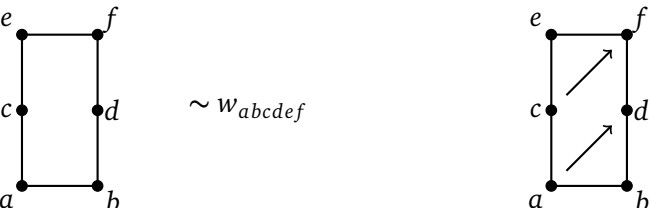

Figure 6: On the left: elementary rectangle for the medium-range interaction-round-a-face models. On the right: the arrows show the direction of the action of the corresponding Lax operator $\check{\mathcal{L}}(u)$.



Figure 7: Transfer matrix for the range 4 models. Now the weights are associated with the elementary rectangles, which give a tiling of the square lattice. The transfer matrix is a segment of the lattice with size $L \times 3$. Periodic boundary conditions are assumed, and only those configurations are allowed which satisfy the constraint for every pair of nearest-neighbour variables.

Now we interpret the Yang-Baxter equation once again as an equation for linear operators. First we define the Lax operator, which now acts on 4 consecutive sites, such that it acts diagonally on the first and last sites. The definition is

$$\check{\mathcal{L}}(u) = \sum_{a,b,c,d,e,f} w_{abcdef}(u) \times \Big( P_e \otimes (|f\rangle\langle c|) \otimes (|d\rangle\langle a|) \otimes P_b \Big). \tag{103}$$

Once again we see that the components of the Lax operator are given by the individual weights, and the Lax operator describes the "diagonal action" on the lattice, see the right hand side of the Fig. 6.

The operator $\check{G}(u,v)$ corresponding to the weights $G(u,v)$ is introduced so that it acts along the horizontal direction, from left to right, on 5 sites, by keeping the first and last site unchanged. With these identifications we obtain the linear equation

$$\check{\mathcal{L}}_{1234}(v)\check{\mathcal{L}}_{3456}(u)\check{G}_{12345}(u,v) = \check{G}_{23456}(u,v)\check{\mathcal{L}}_{1234}(u)\check{\mathcal{L}}_{3456}(v). \tag{104}$$

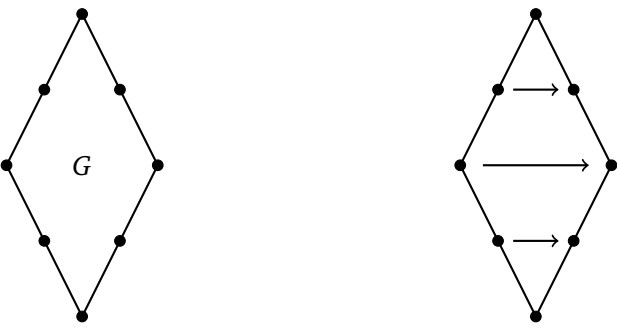

Figure 8: On the left: Depiction of the object $G$, which consists of weights of the allowed configurations of a square of size $2 \times 2$. On the right: when interpreted as a linear operator $\check{G}$ acting on 5 sites, the direction of the action is depicted by the arrows.

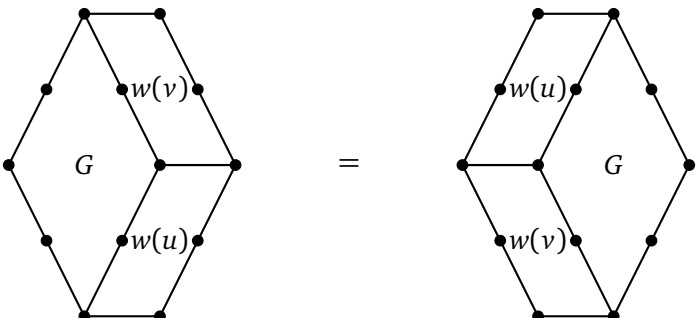

Figure 9: Yang-Baxter relation for the IRF models with range 4 interactions. A summation is understood for the three variables on the inside the hexagonal shape, and it is understood that we only consider configurations satisfying the constraint.

From the transfer matrix we can derive local Hamiltonians in the usual way. We require that $\check{\mathcal{L}}(u = 0)$ becomes the identity operator, and this implies that $t(u)$ becomes the shift operator squared. Taking the first logarithmic derivative we find

$$H = \sum_j h_j, \qquad h_j = \partial_u \check{\mathcal{L}}_{j,j+1,j+2,j+3}(u)|_{u=0}. \tag{105}$$

# 6 Integrable constrained models of range 3

In this section we classify all the time- and space-reflection invariant integrable Hamiltonians of range 3 on the Rydberg-constrained Hilbert space $V_\Pi$. Using (45), our initial Ansatz for the Hamiltonian density is

$$\mathcal{H}_{123} = \alpha P_1 + \beta P_1 P_2 + P_1 \mathcal{O}_2 P_3, \tag{106}$$

where $\mathcal{O}_2 : \mathbb{C}^2 \to \mathbb{C}^2$ is an a priori general operator acting on a single site. Imposing space-/time-reflection invariance, gauge symmetry identities such as (40), and trivial additions of the identity operator in the constrained space we can refine this Ansatz to three independent operators:

$$\mathcal{H}_{123} = h_{12} P_1 X_2 P_3 + h_{11} P_1 P_2 P_3 + h_{22} P_1 N_2 P_3. \tag{107}$$

We form the higher charge density using (24)

$$\mathcal{Q}_{12345} = [\mathcal{H}_{123}, \mathcal{H}_{234} + \mathcal{H}_{345}] + q_{123}. \tag{108}$$

where $q_{123}$ is an a priori arbitrary range 3 operator density. In order to obtain a non-diagonal model we must have $h_{12} \neq 0$, and we normalise $h_{12} = 1$. Forming the total charges in the constrained space[6]

$$Q_2^{(\Pi)} = \Pi\left(\sum_{i=1}^L \mathcal{H}_{i,i+1,i+2}\right)\Pi, \qquad Q_3^{(\Pi)} = \Pi\left(\sum_{i=1}^L \mathcal{Q}_{i,i+1,i+2,i+3,i+4}\right)\Pi, \tag{109}$$

the integrability condition is

$$[Q_2^{(\Pi)}, Q_3^{(\Pi)}] = 0, \tag{110}$$

which in this case reduces simply to the condition $1 + 2h_{11}^2 - h_{11}h_{22} = 0$.

---

[6]A priori $Q_3$ is a range 5 charge, however it is actually range 4 by the definition (45), see (112).

## 6.1 Off-critical golden chain

We find one non-trivial solution to the equation (110):

$$\mathcal{H}_{123} = P_1 X_2 P_3 + z P_1 P_2 P_3 + \left(2z + \frac{1}{z}\right) P_1 N_2 P_3 \,, \tag{111}$$

where $h_{11} := z$ is a free parameter. In particular, the PXP model $\mathcal{H}_{123} = P_1 X_2 P_3$ is not a solution to (110), and so is not integrable. For the model (111) we can take the undetermined range 3 operator $q_{123} = 0$, and the higher charge density can be written explicitly as

$$\mathcal{Q}_{1234} = i P_1 \left(\frac{X_2 Y_3 - Y_2 X_3}{2} + z(P_2 Y_3 - Y_2 P_3)\right) P_4 \,. \tag{112}$$

Up to the addition of a multiple of the identity operator this model can equivalently be written in terms of the density

$$\mathcal{H}'_{123} = P_1 X_2 P_3 + z N_1 P_2 N_3 - \left(z - \frac{1}{z}\right) P_1 N_2 P_3 \,. \tag{113}$$

This model is well-known and has been studied from many different perspectives. It is related to the hard square model of Baxter [51, 79] and coincides with the off-critical RSOS quantum chains discussed in [44]. In the parametrisation (111), there is a critical point at $z = \phi^{5/2}$. At this critical point the model is known as the 'golden chain' [52]. The golden chain is gapless at both ends of the spectrum, and is described by 2d CFTs. Our convention for the Hamiltonian is such that the lowest/highest energy states are described by the CFT with $c = 4/5$ and $c = 7/10$, respectively.

We note that there is another critical point at $z = i\phi^{-5/2}$, where the model reduces to the Lee-Yang chain. This critical model is not Hermitian and describes non-unitary CFTs with central charges $c = -22/5$ and $c = -3/5$. The Hermitian and non-Hermitian critical points are related by the Galois coaction $\phi \to -1/\phi$ [80, 81].

In [44] the Hamiltonian is written in an elliptic parametrisation. To make contact with this Hamiltonian we begin from the Ansatz in a different gauge:

$$\mathcal{H}''_{123} = P_1 X_2 P_3 + \alpha(P_1 P_2 P_3 - P_1 N_2 P_3) + \beta(N_1 P_2 N_3 - P_1 P_2 N_3 - N_1 P_2 N_3) \,. \tag{114}$$

In this case we find the model is integrable on the constrained Hilbert space for

$$(3\beta - 4\alpha)^2 = 25\alpha^2 + 3 \,. \tag{115}$$

The square roots which result from solving for $\beta$ can be resolved by using a theta function parametrisation

$$\alpha = \frac{\vartheta'_1(2\lambda, q)\sqrt{\vartheta_1(\lambda, q)\vartheta_1(2\lambda, q)}}{\vartheta'_1(0, q)\vartheta_1(2\lambda, q)} \,, \tag{116}$$

with $\lambda = \pi/5$. In this parametrisation the critical point occurs in the $q \to 0$ limit. This gauge also makes the Temperly-Lieb structure of the Hamiltonian density manifest.

**Lax operator.** We verify the integrability of the model (111) in our framework by the construction of a Lax operator $\mathcal{L}_{abj}(u)$. We make the Ansatz

$$\mathcal{L}_{abj}(u) = \mathcal{P}_{aj}\mathcal{P}_{bj}(1 + u\mathcal{H}_{abj} + O(u^2)), \tag{117}$$

where in this case we take an auxiliary space $V_A = V_{ab} \simeq \mathbb{C}^2 \otimes \mathbb{C}^2$. The corresponding projected Lax operator (54) is given by

$$\tilde{\mathcal{L}}_{abj}(u) = \Pi_{ab}\mathcal{L}_{abj}(u). \tag{118}$$

We construct the transfer matrix in the constrained space from this Ansatz

$$t^{(\Pi)}(u) = \text{tr}_{ab}[\tilde{\mathcal{L}}_{abL}(u) \cdots \tilde{\mathcal{L}}_{ab1}(u)], \tag{119}$$

and fix the $O(u^2)$ term in (117) by imposing

$$[Q_2^{(\Pi)}, t^{(\Pi)}(u)] = 0. \tag{120}$$

Our solution to this equation is

$$\mathcal{L}_{abj}(u) = \mathcal{P}_{aj}\mathcal{P}_{bj}\left(1 + f(z,u)P_a X_b P_j + uz P_a P_b P_j + \left(\frac{f(z,u)}{z} + uz(2 + uz)\right)P_a N_b P_j\right), \tag{121}$$

where

$$f(z,u) = \frac{1}{2}\left(\sqrt{4u^3z^3 + u^2(z^4 + 6z^2 + 1) + 2u(z^3 + z) + z^2} - uz^2 + u - z\right)$$

$$= u + u^2 z - u^3 + \frac{u^4}{z} + u^5\left(1 - \frac{1}{z^2}\right) + O\left(u^6\right). \tag{122}$$

We stress that (121) contains the same information as the face transfer operators of [44]; it takes a different form because we a working in a different gauge. The transfer matrix (119) constructed from the Lax operator (121) commutes at different values of the spectral parameter

$$[t^{(\Pi)}(u), t^{(\Pi)}(v)] = 0, \tag{123}$$

which is proven by constructing an appropriate $\tilde{R}\tilde{T}\tilde{T}$ relation.

**Constrained integrability and RTT.** As discussed in section 4, the unprojected Lax operator $\mathcal{L}_{abj}(u)$ does not satisfy an RLL relation. This is due to the fact that this model is only integrable on the constrained Hilbert space $V_\Pi$. Therefore, in order to prove the integrability condition (123) we need to solve the R$\tilde{L}\tilde{L}$ relation

$$R_{AB}(u,v)\tilde{\mathcal{L}}_{Aj}(u)\tilde{\mathcal{L}}_{Bj}(v) = \tilde{\mathcal{L}}_{Bj}(v)\tilde{\mathcal{L}}_{Aj}(u)R_{AB}(u,v), \tag{124}$$

where $\tilde{\mathcal{L}}_{Aj}(u)$ is given by (118). We use the auxiliary space multi-indices $A=(a_1 a_2), B=(b_1 b_2)$, which encode $V_A \simeq V_{a_1} \otimes V_{a_2}$ and $V_B \simeq V_{b_1} \otimes V_{b_2}$. We can explicitly solve (124) as a set of linear equations for the entries of $R_{AB}(u,v)$.[7] Due to the projectors in $\tilde{\mathcal{L}}$, there are many entries of the $R$-matrix which vanish in the product $R\tilde{L}\tilde{L}$. We find that this is naturally accounted for by explicitly introducing projectors in $R$ via

$$\tilde{R}_{AB}(u,v) := \Pi_{AB}R_{AB}(u,v), \tag{125}$$

where $\Pi_{AB} := \Pi_{a_1 b_2}\Pi_{a_1 a_2}$ is a projector mixing the auxiliary spaces $V_A$ and $V_B$. With these definitions the projected $R$-matrix $\tilde{R}$ satisfies the Yang-Baxter equation

$$\tilde{R}_{AB}(u,v)\tilde{R}_{AC}(u,w)\tilde{R}_{BC}(v,w) = \tilde{R}_{BC}(v,w)\tilde{R}_{AC}(u,w)\tilde{R}_{AB}(u,v), \tag{126}$$

as well as a modified braiding unitarity

$$\tilde{R}_{AB}(u,v)\tilde{R}_{BA}(v,u) = \alpha(u,v)\Pi_{AB}. \tag{127}$$

The projector $\Pi_{AB}$ is compatible with the projected monodromy matrix

$$\tilde{\Pi}_{BA}\tilde{T}_A(u)\tilde{T}_B(v) = \tilde{T}_A(u)\tilde{T}_B(v), \qquad \tilde{T}_B(v)\tilde{T}_A(u)\tilde{\Pi}_{AB} = \tilde{T}_B(v)\tilde{T}_A(u), \tag{128}$$

where

$$\tilde{T}_A(u) := \tilde{\mathcal{L}}_{AL}(u)\tilde{\mathcal{L}}_{A,L-1}(u) \cdots \tilde{\mathcal{L}}_{A1}(u). \tag{129}$$

Combining these facts, we can use the general arguments in section 4 to prove the integrability condition (123).

---

[7]The expression for $R$ is rather bulky, we prefer to give the corresponding $G$ operator which contains the same information.

**G operator.** We can use the general arguments of section 4.2 to construct a $G$ operator for the model (111). This operator satisfies the constrained GLL relation

$$\check{G}_{234}(u,v)\check{\mathcal{L}}_{123}(u)\check{\mathcal{L}}_{234}(v)\Pi_{\text{open}} = \Pi_{\text{open}}\check{\mathcal{L}}_{123}(v)\check{\mathcal{L}}_{234}(u)\check{G}_{123}(u,v)\,, \tag{130}$$

where $\Pi_{\text{open}} := \Pi_{12}\Pi_{23}\Pi_{34}\Pi_{45}$. Since the Lax matrix is given by (121), this is simply a set of linear equations for the matrix elements of $\check{G}_{ijk}(u,v)$, which admits a solution due to the system being integrable. This solution has the form

$$\begin{aligned}
\tilde{G}_{123}(u,v) &:= \Pi_{12}\Pi_{23}\check{G}_{123}(u,v) \\
&= a(u,v,z)P_1X_2P_3 + b(u,v,z)P_1N_2P_3 + (P_1P_2N_3 + N_1P_2P_3) \\
&\quad + c(u,v,z)P_1P_2P_3 + d(u,v,z)N_1P_2P_3\,.
\end{aligned} \tag{131}$$

We give the full expression for this operator in an auxiliary notebook.

# 7 Integrable constrained models of range 4

We proceed to classify the time- and space-reflection invariant integrable Rydberg-constrained models of range 4. We note that a range 4 Rydberg model has been previously considered in [82], although this case is not integrable. Our initial Ansatz for the Hamiltonian density is

$$\mathcal{H}_{1234} = \alpha P_1 + \beta\ P_1P_2 + P_1\mathcal{O}_2P_3 + P_1\mathcal{O}'_{23}P_4\,. \tag{132}$$

Imposing symmetry, space-reflection invariance on the full Hamiltonian, compatibility with the constraint (33), and relations analogous to (40) and (41) reduces the Ansatz for the Hamiltonian considerably. The reduced Ansatz has six free parameters:

$$\begin{aligned}
\mathcal{H}_{1234} &= h_{11}P_1P_2P_3 + h_{12}P_1X_2P_3 + h_{22}P_1N_2P_3 + g_{11}P_1P_2P_3P_4 \\
&\quad + g_{23}P_1\left(\frac{X_2X_3 + Y_2Y_3}{2}\right)P_4 + g_{12}P_1(X_2P_3 + P_2X_3)P_4\,.
\end{aligned} \tag{133}$$

We form the higher charge density using (24)

$$\mathcal{Q}_{1234567} = [\mathcal{H}_{1234}, \mathcal{H}_{2345} + \mathcal{H}_{3456} + \mathcal{H}_{4567}] + q_{1234}\,, \tag{134}$$

where $q_{1234}$ is an undetermined range 4 operator density. Forming the total charges on the full projected space

$$Q_2^{(\Pi)} = \Pi\left(\sum_{i=1}^{L}\mathcal{H}_{i,i+1,i+2,i+3}\right)\Pi\,, \tag{135}$$

$$Q_3^{(\Pi)} = \Pi\left(\sum_{i=1}^{L}\mathcal{Q}_{i,i+1,i+2,i+3,i+4,i+5,i+6}\right)\Pi\,, \tag{136}$$

the integrability condition is

$$[Q_2^{(\Pi)}, Q_3^{(\Pi)}] = 0\,. \tag{137}$$

New integrable range 4 solutions correspond to those with not all $g_{12}, g_{23}, g_{11}$ being 0, otherwise we recover the off-critical golden chain (111) of the previous section. If $g_{12} = g_{23} = 0$ we find that (137) imposes $g_{11} = 0$, leading to (111). Therefore for new models we necessarily have $\{g_{12}, g_{23}\} \neq \{0,0\}$. For the case $g_{23} = 0, g_{12} \neq 0$ we find no solutions of the integrability condition. For the case $g_{23} \neq 0$ we normalise $g_{23} = 1$ and find two solutions of the integrability condition (137).

### 7.1 Constrained XXZ

The first model we find is equivalent to the constrained XXZ model [57,58]. This corresponds to the Hamiltonian (133) with $h_{12} = g_{11} = g_{12} = 0, g_{23} = 1, h_{11} = a, h_{22} = b$ free:

$$\mathcal{H}_{1234}^{XXZ} = P_1\left(\frac{X_2 X_3 + Y_2 Y_3}{2}\right)P_4 + a P_1 P_2 P_3 + b P_1 N_2 P_3\,. \tag{138}$$

The undetermined range 4 density in (134) can be fixed as

$$q_{1234} = i(a-b)P_1\left(\frac{X_2 Y_3 - Y_2 X_3}{2}\right)P_4\,, \tag{139}$$

and so vanishes for $a = b$. We note that we are always able to choose this undetermined density to be both time- and space-reflection anti-invariant. We see that the model (138) contains and commutes with the particle number operator

$$\mathcal{N}^{(\Pi)} = \Pi\left(\sum_{i=1}^{L} P_i N_{i+1} P_{i+2}\right)\Pi\,. \tag{140}$$

The constrained XXZ model in [57] is written as[8]

$$H_{XXZ}^{(\Pi)} = \frac{1}{2}\Pi\left(\sum_{i=1}^{L} X_i X_{i+1} + Y_i Y_{i+1} + \Delta Z_i Z_{i+2}\right)\Pi\,. \tag{141}$$

This agrees with our model (138) with $a = 2\Delta, b = 4\Delta$, up to the addition of a projected identity operator $9\Delta\Pi^2 = 9\Delta\Pi$. Although this is not manifest, it is true after applying identities such as (40) on the constrained subspace.

**Lax operator.**   We construct the Lax operator for the constrained XXZ model analogously to the range 3 model. For $b = 0$ it is given by

$$\mathcal{L}_{abcj}(u) = \mathcal{P}_{aj}\mathcal{P}_{bj}\mathcal{P}_{cj}\left(1 - u P_a \sigma_b^- \sigma_c^+ P_j - \frac{u}{g(a,u)}P_a \sigma_b^+ \sigma_c^- P_j + \left(1 - \frac{1}{g(a,u)}\right)P_a P_b P_c\right), \tag{142}$$

where

$$g(a,u) = u^2 - au + 1\,. \tag{143}$$

The corresponding projected Lax operator (54) is given by

$$\tilde{\mathcal{L}}_{abcj}(u) = \Pi_{ab}\Pi_{bc}\mathcal{L}_{abcj}(u)\,. \tag{144}$$

**R matrix.**   With this projected Lax operator we can solve the corresponding $\text{R}\tilde{L}\tilde{L}$ relation (62) for the $R$-matrix $R_{AB}(u,v)$. In order to prove integrability we find that we can use the projected $R$-matrix

$$\tilde{R}_{AB}(u,v) = \Pi_{AB} R_{AB}(u,v)\,, \tag{145}$$

with $\Pi_{AB} = \Pi_{a_1 a_2}\Pi_{a_2 a_3}\Pi_{b_2 b_3}\Pi_{b_3 a_1}$. With this definition, the arguments of section 4.1 go through and the integrability condition $[t^{(\Pi)}(u), t^{(\Pi)}(v)] = 0$ can be established.

---

[8]In [57] there is an an overall minus sign and an opposite convention for the Rydberg constraint. We have adjusted their expression to account for this.

**G operator.** The checked version of the Lax operator (142) can be used to construct a $G$ operator, as discussed in section 4.2. In this case the GLL equation reads

$$\check{G}_{23456}(u,v)\check{\mathcal{L}}_{1234}(u)\check{\mathcal{L}}_{3456}(v)\Pi_{\text{open}} = \Pi_{\text{open}}\check{\mathcal{L}}_{1234}(v)\check{\mathcal{L}}_{3456}(u)\check{G}_{12345}(u,v), \tag{146}$$

where $\Pi_{\text{open}} = \Pi_{12}\Pi_{23}\cdots\Pi_{67}$. The corresponding projected $G$ operator is defined by

$$\tilde{G}_{1234}(u,v) := \Pi_{12}\Pi_{23}\Pi_{34}\check{G}_{1234}(u,v). \tag{147}$$

We give the full expression for $\tilde{G}_{1234}(u,v)$ in the auxiliary notebook.

## 7.2 Double golden chain

The other solution to (137) is an apparently new integrable constrained model. Since we have evidence of two critical points related to the golden ratio, we call this the double golden chain. We abbreviate $g_{12} = z$. Then the other coefficients in the Ansatz (133) are fixed as $h_{11} = -1 + z^2$, $h_{12} = -\frac{1+z^2}{z}$, $h_{22} = \frac{1-7z^4+2z^6}{z^2(-1+z^2)}$, $g_{11} = \frac{2z^2}{1-z^2}$, $g_{23} = 1$. If required, the parameter $g_{23}$ can be restored by dimensional analysis. The Hamiltonian reads

$$\mathcal{H}_{1234} = P_1\left(\frac{X_2X_3 + Y_2Y_3}{2} + z(X_2P_3 + P_2X_3) + \frac{2z^2}{1-z^2}P_2P_3\right)P_4 \tag{148}$$
$$+ P_1\left(-\frac{1+z^2}{z}X_2 + (z^2-1)P_2 + \frac{1-7z^4+2z^6}{z^2(-1+z^2)}N_2\right)P_3.$$

The undetermined range 4 operator in (134) can be chosen as

$$q_{1234} = -\frac{iz^2(z^2-3)}{z^2-1}P_1\left(\frac{X_2Y_3 - Y_2X_3}{2}\right)P_4 \tag{149}$$
$$+ iz(Y_1P_2N_3 - N_1P_2Y_3 + N_1P_2Y_3N_4 - N_1Y_2P_3N_4 + N_1N_2P_3Y_4 - Y_1P_2N_3N_4).$$

We note that in this gauge we cannot pick $q_{1234} = 0$ for any non-zero $z$. In a gauge of purely length four operators this Hamiltonian admits a hyperbolic parametrisation

$$\mathcal{H}_{1234} = P\left(\frac{XX + YY}{2}\right)P + \sinh\eta(PXPP + PPXP) + \cosh\eta(PXPN + NPXP) \tag{150}$$
$$+ \alpha(NPPP + PPPN) + (\alpha - \beta)(PNPP + PPNP) + \gamma(PNPN + NPNP),$$

where $\alpha = \frac{1}{2}(1 - \coth\eta)$, $\beta = 2\sinh^2\eta$, $\gamma = -e^\eta\sinh\eta$, and we omitted indices since there is no ambiguity.

**Lax operator.** While we were not able the compute the full form of the Lax operator corresponding to (148) analytically, we have computed it perturbatively to high orders, which is strong evidence for its integrability. Using the Hamiltonian (148) we make the initial Ansatz

$$\mathcal{L}_{abcj}(u) = \mathcal{P}_{aj}\mathcal{P}_{bj}\mathcal{P}_{cj}\left(1 + u\mathcal{H}_{abcj} + \sum_{k=2}^{\infty}u^k(\mathcal{O}_k)_{abcj}\right). \tag{151}$$

Using the Ansatz (151), we form the transfer matrix

$$t^{(\Pi)}(u) = \text{tr}_{abc}\left[\tilde{\mathcal{L}}_{abcL}(u)\cdots\tilde{\mathcal{L}}_{abc1}(u)\right]. \tag{152}$$

The operators $(\mathcal{O}_k)_{abcj}$ can be fixed by solving the condition

$$[Q_2^{(\Pi)}, t^{(\Pi)}(u)] = 0, \tag{153}$$

at each order in $u$. We find that each $\mathcal{O}_k$ admits the expansion

$$
\begin{aligned}
(\mathcal{O}_k)_{1234} = {} & a_{1,k}(z)P_1 P_2 \sigma_3^- P_4 + a_{2,k}(z)P_1 \sigma_2^- P_3 P_4 + a_{3,k}(z)P_1 P_2 P_3 N_4 \\
& + a_{4,k}(z)P_1 \sigma_2^- P_3 N_4 + a_{5,k}(z)P_1 \sigma_2^- \sigma_3^+ P_4 + a_{6,k}(z)P_1 \sigma_2^- P_3 P_4 \\
& + a_{7,k}(z)P_1 N_2 P_3 P_4 + a_{8,k}(z)P_1 \sigma_2^+ P_3 N_4 + a_{9,k}(z)P_1 N_2 P_3 N_4 \,.
\end{aligned}
\tag{154}
$$

To verify the integrability of the model, we have computed $a_{i,k}$ for $k = 2, \ldots, 10$ as functions of the coupling $z$.[9] For example, the coefficients $a_{i,2}$ are given by

$$
\begin{aligned}
& a_{1,2} = z \,, && a_{2,2} = -\frac{z(z^2 - 3)}{z^2 - 1} \,, && a_{3,2} = \frac{z^4 + z^2}{z^2 - 1} \,, \\
& a_{4,2} = \frac{-z^5 + z^3 + 2z}{z^2 - 1} \,, && a_{5,2} = \frac{z^2(z^2 - 3)}{z^2 - 1} \,, && a_{6,2} = \frac{2z}{z^2 - 1} \,, \\
& a_{7,2} = \frac{z^8 - 8z^6 + 15z^4 - 4}{(z^2 - 1)^2} \,, && a_{8,2} = \frac{-z^5 + z^3 + 2z}{z^2 - 1} \,, && a_{9,2} = \frac{(z^2 - 3)(z^6 - 4z^4 + 1)}{(z^2 - 1)^2} \,.
\end{aligned}
\tag{155}
$$

These Lax coefficients encode the information of the operator $Q_3^{(\Pi)}$, which commutes with $Q_2^{(\Pi)}$. In general, the coefficients $a_{i,k}$ encode the information of the higher commuting charge $Q_{k-1}^{(\Pi)}$.

# 8 Properties of the double golden chain

In this section we will discuss some of the properties of the double golden chain. In particular, we highlight its relation to the golden chain, its critical points and some of its symmetries. We also discuss this model in the context of a deformation from the PXP model. To this end, we renormalise our Hamiltonian density (148)

$$
\mathcal{H}'_{1234} = A\Big[ P_1 \Big( z^2(z^2 - 1)\frac{X_2 X_3 + Y_2 Y_3}{2} + z^3(z^2 - 1)(X_2 P_3 + P_2 X_3) - 2z^4 P_2 P_3 \Big) P_4 \tag{156}
$$
$$
+ P_1 \Big( -z(z^2 - 1)(z^2 + 1)X_2 + (z^2 - 1)^2 z^2 P_2 + (1 - 7z^4 + 2z^6)N_2 \Big) P_3 \Big],
$$

with
$$
A^{-1} = \sqrt{2}\sqrt{5z^{12} - 32z^{10} + 60z^8 - 4z^6 - 14z^4 + 2z^2 + 1} \,,
\tag{157}
$$

so that $\mathcal{H}'_{1234}$ has standard norm 1. This will provide a convenient normalisation when we will study the spectrum and gap in the remainder of this section.

## 8.1 Gap analysis

**Diagonal points.** We first note that there are three special points where the Hamiltonian becomes diagonal. It is useful to study these simple points in parameter space, in order to get an idea about the nature of the various phases of the model. We will see that the three diagonal points have quite different properties, including a spatial periodicity going from $\mathbb{Z}_2$ through $\mathbb{Z}_4$ to $\mathbb{Z}_3$ order. This phenomenon already suggests the existence of phase transitions between the different phases (for a similar phenomenon in the changes in the spatial periodicity see [51]).

---

[9]In principle we can generate the coefficients $a_{i,k}$ to any order we want, it is just a matter of computation. At each order in $u$ the condition (153) is a set of linear equations for these coefficients.

We find the following diagonal points:

- At $z = 0$ the Hamiltonian reduces to

$$\mathcal{H}'_{1234} = \sqrt{2} P_1 N_2 P_3 \,, \tag{158}$$

which is proportional to the number operator that counts the number of excitations in a state. Because of the constraint, we see that this is different for odd- and even-length states. We find that the spectrum is of the form $E = \sqrt{2}n$, where

$$n \in \{0, \dots, L/2\}\,, \qquad L \text{ even},$$
$$n \in \{0, \dots, (1-L)/2\}\,, \quad L \text{ odd}.$$

In both cases the ground state is simply given by the ferromagnetic state with all spins up. For even $L$, the anti-ground state is two-fold degenerate and is given by a linear combination of the two states $|\uparrow\downarrow\uparrow \dots\rangle$ and $|\downarrow\uparrow\downarrow \dots\rangle$. For odd $L$ there is an $L$-fold degeneracy and the anti-ground state is given by $|\uparrow\uparrow\downarrow\uparrow\downarrow \dots\rangle$, and the remaining states are obtained by acting on this state with the shift operator.

- At $z = 1$ the Hamiltonian takes a more complicated form

$$\mathcal{H}'_{1234} = -\frac{1}{3} P_1 P_2 P_3 P_4 - \frac{2}{3} P_1 N_2 P_3 \,. \tag{159}$$

At this point, the Hamiltonian density is of range 4, and the nature of the spectrum depends on $a \equiv L \bmod 4$. We write $L = 4k + a$. We can write the eigenvalues at this point as $E = n/6$ with

$$n = \{L, \dots, 2L\}\,, \qquad a = 0\,,$$
$$n = \{L+1, \dots, 2L\}\,, \qquad a = 1\,,$$
$$n = \{L+2, \dots, 2L\}\,, \qquad a = 2\,,$$
$$n = \{L+1, \dots, 2L\}\,, \qquad a = 3\,.$$

The ground state and anti-ground state structure is more involved here as well. When $a = 1, 3$ the ground state is non-degenerate and the anti-ground state is $L$-fold degenerate. When $a = 0$, we find that the ground state is three-fold degenerate while the anti-ground state is four-fold degenerate. Finally, when $a = 2$ we find that the ground state is again three-fold degenerate, but the anti-ground state is $(2k+3)(2k+1)$ degenerate.

- At $z = \infty$ we get

$$\mathcal{H}'_{1234} = \frac{1}{\sqrt{10}} P_1 (P_2 + 2N_2) P_3 \,. \tag{160}$$

This Hamiltonian is a range 3 operator and the nature of the spectrum depends on $b \equiv L \bmod 3$, so we write $L = 4k + b$. The energies take values $E = \frac{n}{\sqrt{10}}$, where $n \in \{2k+b, \dots, L\}$. However, the degeneracies differ depending on $b$. For $b = 0, 1, 2$ we find a degeneracy of 3, $L$, and $\frac{1}{2}(k+3)(3k+2)$ respectively. For the anti-ground state we note that it is unique for odd $L$ and three-fold degenerate for even $L$.

**Spectrum.** There are two interesting ranges for our interaction parameter $z$. By construction, when $z$ is real, our Hamiltonian is Hermitian and we have a real spectrum. Due to the fact that the ground state has different degeneracies between the special points $z = 0, 1, \infty$ we expect phase transitions between those points. We will expand on this in the next section when we study the gap of this model.

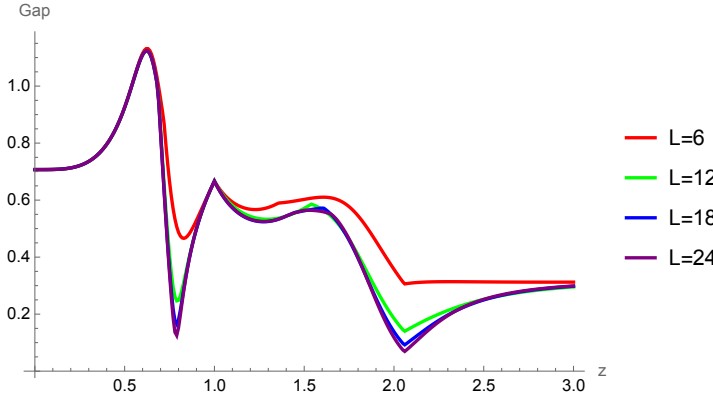

Figure 10: The energy gap for $L = 6, 12, 18, 24$ for different values of the coupling constant $z$. The dips in the plot correspond to the critical points at $z = \phi^{-1/2}$ and $z = \phi^{3/2}$ respectively.

However, when $z = i\zeta$ is purely imaginary we find that the Hamiltonian is pseudo-Hermitian when $|\zeta| \leq \phi^{-3/2}$ for any $L$. This means that $H$ and $H^\dagger$ are related by a similarity transformation, which implies that the spectrum is purely real as well. When $|\zeta| > \phi^{-3/2}$ then the spectrum is still real when $L$ is odd. For even $L$ the eigenvalues $E$ with $\text{Im}(E) \neq 0$ appear in complex conjugate pairs.

Finally, we note that at the point $z = \phi^{3/2}$ the spectrum of the double golden chain coincides with the spectrum of the golden chain for odd $L$. For even $L$ at this point there is a more complicated relation. We find that part of the spectrum is given by combinations of eigenvalues of two golden chains with length $L/2$. We will give some motivation for these observations below when we discuss Temperley-Lieb representations.

**Gap and critical points.** Due to the differences in the degeneracies of the ground state at the points $z = 0, 1, \infty$ we expect the model to exhibit two phase transitions: one for $0 < z_1 < 1$ and one for $z_2 > 1$. To find the critical parameters, we look at the energy gap of the model. We have computed the gap for $L = 6, 12, 18, 24$, see Figure 10. We plot the gap between the ground state and the third excited state. This is due to the fact that after crossing some critical points the energy of the lower excited states coincides with the ground state energy. In other words, as argued above, we see that in some regions the degeneracy of the ground state increases because of this.

From our numerics we indeed find that there are two critical points where the gap seems to vanish in the large $L$ limit. Numerical evidence suggests that these points are given by $z_1 = \phi^{-1/2}$ and $z_2 = \phi^{3/2}$. At these critical points we find the expected $1/L$ behaviour of the gap, see Figure 11. If we slightly deviate from these critical values, the $1/L$ behaviour seems to disappear.

## 8.2 Temperley-Lieb algebra

At the critical point $z = \phi^{3/2}$, we can find a gauge where the Hamiltonian (148) has a Temperley-Lieb-like structure. The generators satisfy a modified Temperley-Lieb algebra, which is actually two algebras intertwined, as we explain below. We are not aware of any other example for such an algebra.

In the appropriate gauge the total Hamiltonian takes the form

$$Q_2^{(\Pi)} = \Pi \left( \sum_{i=1}^{L} e_i \right) \Pi, \tag{161}$$

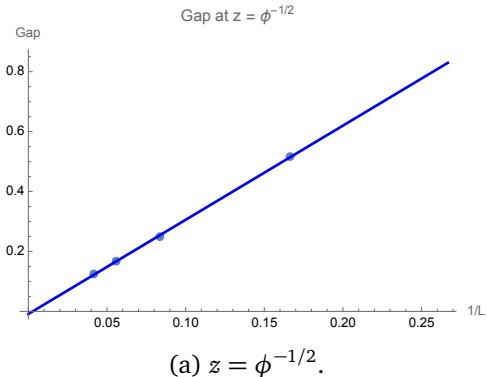

(a) $z = \phi^{-1/2}$.

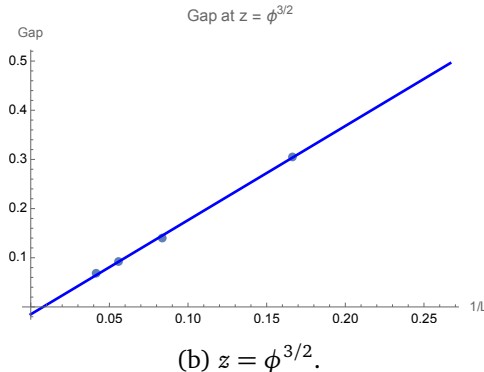

(b) $z = \phi^{3/2}$.

Figure 11: The difference between the energy of the first excited state and the ground state at $z = \phi^{-1/2}$ and $z = \phi^{3/2}$ as a function of $1/L$. The dots are the values at $L = 6, 12, 18, 24$ and the line is the best linear fit. We see a very clean linear dependence on $1/L$.

where

$$e_i = P\left(\frac{XX + YY}{2}\right)P + \phi^{-1/2}(PXPP + PPXP) - \phi^{1/2}(PXPN + NPXP)$$
$$+ \phi(NPPP + PPPN) + (PNPP + PPNP + PNPN + NPNP) + \phi^{-1}PPPP\,. \quad (162)$$

The operators $e_i$ satisfy the algebra

$$[e_i, e_{i\pm 1}] = 0\,, \quad (163)$$

$$e_i e_{i\pm 2} e_i = e_i\,, \quad (164)$$

$$e_i^2 = \phi\, e_i\,, \quad (165)$$

$$[e_i, e_j] = 0\,, \qquad |i - j| > 2\,. \quad (166)$$

We note that this model has the same Temperley-Lieb parameter $\phi$ as the golden chain, however the algebra is slightly different. All the operators commute except those separated by exactly two sites. Hence the Temperley-Lieb algebra we find here is isomorphic to that of the golden chain, but is only represented on even or odd sites. If we have periodic boundary conditions and the length of the chain is odd, then the algebra becomes actually identical to that of the golden chain. This leads to the interesting observation that for odd lengths, the spectrum at $z = \phi^{3/2}$ completely coincides with the spectrum of the golden chain. However, for even lengths $L = 2n$ (or for open boundary conditions) we find two separate copies of the Temperley-Lieb algebra, which are nevertheless intertwined through the representation itself. Numerical data shows that many of the energy levels are actually sums of the energy levels from the golden chain with half the length ($L' = n$), but not all energy levels are reproduced this way. The exact relation for even lengths remains unclear, and it would be interesting to study this.

The intertwining of two copies of the Temperley-Lieb algebra suggests that the scaling limit of the spin chain is described by two copies of the corresponding CFTs of the golden chain. For the ground state (lowest energy) this is the minimal model with $c = 4/5$, whereas for the anti-ground state (highest energy) it is the minimal model with $c = 7/10$.

Using the Galois coaction [81] we note there is an imaginary critical point $z = i\phi^{-3/2}$ where the Hamiltonian is also Temperley-Lieb-like. At this point the Hamiltonian is non-Hermitian and can be written as

$$Q_2^{(\Pi)} = \Pi\left(\sum_{i=1}^{L} f_i\right)\Pi\,, \quad (167)$$

where

$$f_i = P\left(\frac{XX+YY}{2}\right)P + i\phi^{-1/2}(PXPP+PPXP) + i\phi^{1/2}(PXPN+NPXP)$$
$$- \phi^{-1}(NPPP+PPPN) + (PNPP+PPNP+PNPN+NPNP) - \phi PPPP. \quad (168)$$

In this case the operators $f_i$ satisfy

$$[f_i, f_{i\pm 1}] = 0, \quad (169)$$
$$f_i f_{i\pm 2} f_i = f_i, \quad (170)$$
$$f_i^2 = \phi^{-1} f_i, \quad (171)$$
$$f_i f_j - f_j f_i = 0, \qquad |i-j| > 2. \quad (172)$$

This model is the analogue of the Lee-Yang chain; the spectral equivalence is similar to that of above. These are the *only* points where the Hamiltonian can be brought into a Temperley-Lieb-like form. In particular, we were unable to identify such an algebraic structure at the other real critical point $z = \phi^{-1/2}$.

We mention one more interesting point, at $z = i$. At this point the Hamiltonian can be written in a form which is somewhat similar to the Temperley-Lieb algebra. In particular we find that

$$Q_2^{(\Pi)} = \Pi\left(\sum_{i=1}^{L} g_i\right)\Pi, \quad (173)$$

where

$$g_i = P\left(\frac{XX+YY}{2}\right)P + i(PXPP+PPXP) + (PNPP+PPNP-PPPP). \quad (174)$$

The operators $g_i$ satisfy the non-standard algebra

$$g_i g_{i\pm 1} = 0, \quad (175)$$
$$g_i g_{i\pm 2} g_i = 0, \quad (176)$$
$$g_i^2 = g_i, \quad (177)$$
$$g_i g_j - g_j g_i = 0, \qquad |i-j| > 2. \quad (178)$$

It is also worthwhile to notice that the specturm at $z = i$ is degenerate. This is because the operators $g_i$ enjoy an enhanced symmetry, whereby they commute with the diagonal operator $PNPP + PPNP + PPPP$. Moreover it can be checked that this is the only value of $z$ where the Hamiltonian commutes with a diagonal operator.

## 8.3 Deformation of PXP

We investigate the double golden chain (148) as a deformation of the PXP model. In [70] a range 4 Hamiltonian was found near PXP with strong signatures of integrability based on level statistics:

$$\mathcal{H}_{1234} = -P_1 X_2 P_3 + \alpha(P_1 X_2 P_3 Z_4 + Z_4 P_2 X_3 P_4), \quad (179)$$

where $\alpha \sim 0.02$. We note that there are no values of $\alpha$ for which the model is exactly integrable, based on the criterion (137). However, we can start from a complete range 4 Ansatz which includes the operators in (179). In this way we find the double golden chain in a basis which naturally includes (179):

$$\mathcal{H}_{1234}^{\mathrm{DGC}} = -P_1 X_2 P_3 + \alpha(P_1 X_2 P_3 Z_4 + Z_4 P_2 X_3 P_4) + \sqrt{\frac{\alpha}{2}} P_1 (X_2 X_3 + Y_2 Y_3) P_4 \quad (180)$$

$$+ \frac{4\sqrt{2}\alpha^{3/2}}{1-2\alpha} P_1 P_2 P_3 P_4 + \sqrt{2}\alpha(2\alpha-1) P_1 P_2 P_3 + \frac{16\alpha^3 - 28\alpha^2 + 1}{\sqrt{2\alpha}(2\alpha-1)} P_1 N_2 P_3.$$

We do not find any special behaviour for this Hamiltonian near the point $\alpha \sim 0.02$. In particular, the hopping term $P_1(X_2X_3 + Y_2Y_3)P_4$ and diagonal terms remain large compared to $\alpha$ at this point. These operators do not commute with the operator $\Pi_{i=1}^L Z_i$ on the whole chain; their addition to the PXP model rapidly leads to thermalisation. We therefore exclude the double golden chain (180) as a candidate for the integrable model close to PXP.

# 9 Conclusions and outlook

In this paper we began a systematic study of integrable models on Rydberg-constrained Hilbert spaces. Using a modification of the Reshetikhin condition we formulated an integrability condition on such Hilbert spaces, and fully classified all time- and space- reflection invariant integrable models up to range 4. We discussed how to modify the RLL framework of integrability to include such integrable models, and the relation of this framework to the RSOS models of statistical physics. One of our key results was the discovery of the double golden chain: a new range 4 integrable model with many interesting properties.

There are several directions for future study. One important task is to push the classification to higher-range. While the Reshetikhin condition gives a (conjecturally) complete set of conditions for a given Hamiltonian density to be integrable, in practice this set of equations is strongly coupled and difficult to solve exactly as the number of parameters grows. It is possible that there is a change of variables/different operator basis which exposes the structure of the equations and renders them easier to solve. We also did not yet attempt any algebraic methods for solving these systems of polynomial equations, for example Gröbner bases. An important first step would be to complete the classification of range 5 integrable models, where there are 14 free parameters.

There are several applications for new integrable models on the Rydberg-constrained Hilbert space. One key goal is to settle the question whether there exists an integrable model proximate to the PXP model, and if so to find it analytically. While we do not have any evidence that this hypothetical model is related to any the new integrable models in this paper, we cannot exclude the possibility that it corresponds to a not-yet-classified model at range 5 or higher. It is also possible that the parent integrable model is long-range: the methods of this paper do not apply to this case, and new tools would need to be developed to study this. Integrable models of Rydberg atoms have also proven to possess 'golden' critical points corresponding to CFTs. It would be interesting to investigate if this pattern persists at higher ranges.

While the results of this paper were for the Rydberg constraint, they can be tailored easily to other local constraints. This allows for the possibility of finding new integrable models with a given constraint. For a two-dimensional local Hilbert space, the only possibility is to increase the range of the constraint. A higher-range generalisation of the Rydberg constraint is requiring a distance of at least $D$ between down spins (the Rydberg constraint corresponds to $D = 2$). Such models were considered in [57], and constitute generalisations of the constrained XXZ model discussed in this paper. For local Hilbert space dimensions 3 and higher, there is a large number of possible local constraints, which could lead to many new integrable models. Although it is unclear if other constraints have any experimental relevance, such new models may be interesting from a theoretical standpoint.

It would also be interesting to study the double golden chain in more detail. While we computed the Lax operator of this model perturbatively, we were not able to compute it analytically to all orders in the spectral parameter. Identifying the function class which appears in the Lax operator would lead to a deeper understanding of the model. While we identified the critical points of the model, it still remains to identify the precise CFTs to which these cor-

respond. At the critical point $z = \phi^{3/2}$ we demonstrated a doubled Temperley-Lieb algebra. This explains the coincidence of the spectrum with the that of the golden chain at odd lengths. The relation between the spectra at even lengths appears more intricate, however, and should be studied further. Based on the doubled Temperley-Lieb algebra, we conjectured that this point is described by two copies of the corresponding CFT's of the golden chain, with central charges $c = 7/10$ and $c = 4/5$. The critical point at $z = \phi^{-1/2}$ is more mysterious; it would be interesting to investigate if there are any similar algebraic structures emerging at this point.

Finally, while integrability of a model often leads to its exact solvability by a Bethe Ansatz approach, this has not been demonstrated for medium-range models to the same extent as for the nearest-neighbour case. Although some models with spin conservation, such as the constrained XXZ model [57], can be treated with a coordinate Bethe Ansatz, the algebraic counterpart is still missing. It is an important question for future research, whether the Lax operators and transfer matrices in the medium-range RLL formalism can be used to to construct an algebraic Bethe Ansatz, both for constrained and unconstrained models.

## Acknowledgments

We are thankful for motivating discussions with Maksym Serbyn and Marko Ljubotina. We also thank Tristan McLoughlin for useful discussions. B.P. acknowledges collaboration on early stages of this project with Tamás Gombor.

**Funding information** MdL was supported in part by SFI and the Royal Society for funding under grants UF160578, RGF\R1\181011, RGF\8EA\180167 and RF\ERE\210373. MdL is also supported by ERC-2022-CoG - FAIM 101088193. LC was supported by RF\ERE\210373. B.P. was supported by the NKFIH excellence grant TKP2021_NKTA_64.

## A  Partial classification for range 5

In this appendix we provide a partial classification of time- and space-reflection invariant integrable models at range 5. Imposing compatibility with the constraint, time- and space-reflection invariance, and identities like (40) we arrive at our range 5 Ansatz with 14 free parameters:

$$
\begin{aligned}
\mathcal{H}_{12345} = {} & h_{11}P_1P_2P_3 + h_{12}P_1X_2P_3 + h_{22}P_1N_2P_3 + g_{11}P_1P_2P_3P_4 \\
& + g_{23}P_1\left(\frac{X_2X_3 + Y_2Y_3}{2}\right)P_4 + g_{12}P_1(X_2P_3 + P_2X_3)P_4 \\
& + f_{11}P_1P_2P_3P_4P_5 + f_{12}(P_1X_2P_3P_4P_5 + P_1P_2P_3X_4P_5) \\
& + f_{23}P_1\left(\frac{X_2X_3P_4 + Y_2Y_3P_4 + P_1X_2X_3 + P_1Y_2Y_3}{2}\right)P_5 \\
& + f_{13}P_1P_2X_3P_4P_5 + f_{16}P_1\left(\frac{X_2P_3X_4 - Y_2P_3Y_4}{2}\right)P_5 \\
& + f_{25}P_1\left(\frac{X_2P_3X_4 + Y_2P_3Y_4}{2}\right)P_5 + f_{33}P_1P_2N_3P_4P_5 \\
& + f_{36}P_1\left(\frac{X_2X_3X_4 - Y_2X_3Y_4 + X_1Y_2Y_3 + Y_1Y_2X_3}{4}\right)P_5 .
\end{aligned}
\tag{A.1}
$$

From the Hamiltonian density (A.1) we form the higher charge density using (24):

$$\mathcal{Q}_{123456789} = [\mathcal{H}_{12345}, \mathcal{H}_{23456} + \mathcal{H}_{34567} + \mathcal{H}_{45678} + \mathcal{H}_{56789}] + q_{12345}, \tag{A.2}$$

where $q_{12345}$ is an undetermined range 5 operator density. Forming the total charges on the projected space

$$Q_2^{(\Pi)} = \Pi \left( \sum_{i=1}^{L} \mathcal{H}_{i,i+1,i+2,i+3,i+4} \right) \Pi, \tag{A.3}$$

$$Q_3^{(\Pi)} = \Pi \left( \sum_{i=1}^{L} \mathcal{Q}_{i,i+1,i+2,i+3,i+4,i+5,i+6,i+7,i+8} \right) \Pi, \tag{A.4}$$

the integrability condition is

$$[Q_2^{(\Pi)}, Q_3^{(\Pi)}] = 0. \tag{A.5}$$

The condition (A.5) corresponds to a large non-linear system of coupled equations for the 14 couplings in (A.1). While we are not yet able to solve these equations fully, it is possible to classify all the solutions in various special cases. In this paper we only present integrable Hamiltonians; we leave the analysis of corresponding Lax operators for future work.

**Spin conservation.**    We are able to fully classify integrable range 5 models with spin conservation, i.e. those commuting with the particle number operator

$$\mathcal{N}^{(\Pi)} = \Pi \left[ \sum_{i=1}^{L} P_i N_{i+1} P_{i+2} \right] \Pi. \tag{A.6}$$

These models correspond to the Ansatz (A.1) with $h_{12} = g_{12} = f_{12} = f_{13} = f_{16} = f_{36} = 0$. In particular, the only allowed interaction terms are $P(XX + YY)P, P(XPX + YPY)P$, and $P(XXP + YYP + PXX + PYY)P$. In this case we find two solutions to the integrability condition (A.5). The first model we find is

$$\mathcal{H}_A = z P_1 \left( \frac{X_2 P_3 X_4 + Y_2 P_3 Y_4}{2} \right) P_5 + w \left( P_1 P_2 P_3 - P_1 P_2 P_3 P_4 \right).$$

A potentially nicer rewriting is

$$\mathcal{H}_A = z P_1 \left( \frac{X_2 P_3 X_4 + Y_2 P_3 Y_4}{2} \right) P_5 + w P_1 \left( \frac{N_2 P_3 P_4 + P_2 P_3 N_4}{2} \right) P_5.$$

This model exhibits many interesting properties such as Hilbert space fragmentation. The second model we find in this subclass is

$$\mathcal{H}_B = z P_1 \left( \frac{X_2 X_3 P_4 + P_2 X_3 X_4 + Y_2 Y_3 P_4 + P_2 Y_3 Y_4}{2} \right) P_5 - z P_1 \left( \frac{X_2 X_3 + Y_2 Y_3}{2} \right) P_4 + w P_1 P_2 P_3.$$

This new range 5 integrable Rydberg-constrained model exhibits nearest-neighbour hopping.

$f_{16} = f_{36} = 0, \; f_{25} \equiv z \neq 0.$    In this case we find 2 solutions of the integrability condition (A.5). The first model is $\mathcal{H}_A$ from above. The second model we find in this subclass is a two-parameter model:

$$\mathcal{H}_C = z P_1 \left( \frac{X_2 P_3 X_4 + Y_2 P_3 Y_4}{2} \right) P_5 + y P_1 (X_2 P_3 + P_2 X_3) P_4 - 2y P_1 X_2 P_3$$
$$+ P_1 \left( \frac{2y^2}{z} P_2 + \left( \frac{4y^2}{z} - 2z \right) N_2 \right) P_3. \tag{A.7}$$

This is an apparently new constrained integrable model which contains a spin-conserving hopping term of range 2 as well as explicit spin-violating interactions. We note that although this model contains a $PXP$ interaction, it is not continuously connected to the $PXP$ model.

$f_{16} = f_{36} = f_{25} = f_{12} = f_{23} = 0,\ f_{13} \equiv z \neq 0.$   Also solvable is the case where the only non-zero range 5 interaction corresponds to the coupling $f_{13} \equiv z$. In this case there are 2 solutions to (A.5). The first is

$$\mathcal{H}_D = zP_1P_2X_3P_4P_5 - zP_1\left(\frac{X_2P_3 + P_2X_3}{2}\right)P_4 + qP_1P_2P_3P_4 - qP_1(P_2P_3P_4 + P_2N_3P_4)P_5. \quad \text{(A.8)}$$

This model can equivalently be rewritten as

$$\mathcal{H}_D = z(P_1P_2X_3P_4N_5 + N_1P_2X_3P_4P_5) + qP_1P_2P_3P_4 - qP_1(P_2P_3P_4 + P_2N_3P_4)P_5, \quad \text{(A.9)}$$

and commutes with the $U(1)$ charge

$$\Pi\left[\sum_{i=1}^{L} P_iP_{i+1}P_{i+2} + 2P_iN_{i+1}P_{i+2}\right]\Pi. \quad \text{(A.10)}$$

The second model we find is

$$\mathcal{H}_E = z(P_1P_2X_3P_4N_5 + N_1P_2X_3P_4P_5) - \frac{z}{2}P_1X_2P_3 + P_1(wP_2 + \left(\frac{z^2}{4w} + 2w\right)N_2)P_3.$$

**Higher charge of the off-critical golden chain.**   We mention one other solution of the range 5 integrability condition (A.5). This corresponds to the Hamiltonian (A.1) with a normalised $f_{16} = 1$, and $f_{13} \equiv z$ and $h_{12} \equiv y$ as free parameters. The other non-zero couplings are fixed via

$$\begin{aligned} g_{12} &= -\frac{1+z^2}{2z}, & g_{23} &= -\frac{3}{2} - \frac{1}{2z^2}, & f_{23} &= -f_{25} = -g_{11} = 1, \\ f_{36} &= \frac{1}{z}, & h_{11} &= -2 + yz, & h_{22} &= -6 + y\left(2z + \frac{1}{z}\right). \end{aligned} \quad \text{(A.11)}$$

If required, $f_{16}$ can be restored by dimensional analysis. Although this model appears mysterious at first, we find that it commutes with the range 3 integrable model (111) on the constrained subspace. Therefore, it is related to the higher charge $Q_4^{(\Pi)}$ of this model, and so does not constitute a new range 5 integrable Rydberg-constrained model. We note that while there are two parameters for this model, $z$ and $y$, the coupling $y$ simply corresponding to a range 5 embedding of the off-critical golden chain $Q_2^{(\Pi)}$, and the parameter $z$ contains the non-trivial range 5 information corresponding to the higher charge $Q_4^{(\Pi)}$.

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
