# Peer review of "Integrable models on Rydberg atom chains"

_SciPost Physics, doi:SciPost Phys. 18, 139 (2025)_

## Round 2 · Referee Report · Anonymous (Referee 1) · 2024-11-11

Strengths

  1. Paper addresses a timely and interesting question at the intersection of two fields: integrability and constrained models.
  2. Paper presents details in the clear form, and results can be readily used by the community.

Weaknesses

  1. Paper is written in a somewhat overly technical way. While it is impossible to avoid technical language in a field of integrability, I still believe that some things can be explained more intuitively/deciphered.
  2. In some places it would be nice to expand the paper with the examples.

Report

The paper addresses a timely and interesting question at the intersection of two fields: integrability and constrained models. The main results of the paper are new families of integrable constrained models, beyond those known in the literature. I find these results highly interesting and non-trivial, as integrable models often allow unique insights into the physics due to their analytical solvability.

I am not an expert in integrability, but some of the checks confirm self-consistency of results. In particular, the dependence of Eq. (6) on parameter z is consistent with spectral reflection property (existence of operator C = prod sigma^z that changes sign of some terms in the Hamiltonian, thus mapping z->-z effectively).

While I am happy to recommend the publication without any changes, I invite authors to address some of my comments to improve the readability and presentation of the paper.

Requested changes

General comments:

  1. It would be nice to comment if models found by authors have many-body chiral symmetry, i.e. if there exists points where some operator C satisfies {C,H} =CH+HC=0.
  2. When deriving families of models, it would be interesting to see the flavor of algebraic equations derived for free parameters.
  3. Would be nice to have some discussion of connection between list of bullet points on page 8 and 16 (if it exists).
  4. Section 6 has only one subsection. Please restructure.
  5. Another interesting speculative point for discussion would be the possibility of disordered (with correlated disorder) constrained integrable models. Would the author's method be capable of finding such models?

Most of changes are minor and would improve readability:

  1. The very first sentence in the paper would benefit from rephrasing: it is too vague.
  2. Same first paragraph term "algebraic origin" is not quite clear
  3. Some terms would be good to unpack for a non-expert audience, for instance algebraic Bethe ansatz, Reshetikhin conditions, RLL relation,... by just explaining their essence/flavor. This will improve the readability for non-experts in integrability.
  4. References on Page 3 [32,...] are grouped into strange groups. Join them or explain.
  5. Footnote 1 should be after the dot.
  6. Explicitly mark special points that are studied in Fig 11 on the x-axis of Fig. 10

Recommendation

Publish (surpasses expectations and criteria for this Journal; among top 10%)

---

## Round 2 · Referee Report · Anonymous (Referee 2) · 2025-3-3

Strengths

  1. The paper provides the first systematic approach to construct integrable models in restricted Hilbert spaces.
  2. They constructed a new family of integrable models and provided a first investigation on their properties.
  3. The paper is well written and well structured, with a clear introduction and an interesting discussion presented in the conclusions. Appropriate references are also provided.

Weaknesses

None that I can see.

Report

The paper provides a novel link between different research areas by connecting models in restricted Hilbert spaces, integrability and medium range spin chains. This is done by developing a new systematic approach to construct integrable models in restricted Hilbert spaces. This new construction builds on recent works on medium range spin chains.

The authors used the method to classify integrable models with Rydberg constraint, for both range 3 and range 4 chains, and partially classified those of range 5. A new family of integrable models is presented and a preliminary analysis of various regimes of this model is presented.

The paper is clearly written, with detailed explanations, and relevant references. The introduction presents a very clear description of the state of the art in the topic. The conclusions clearly summarise the work and identify open problems and important next steps. The most interesting of those in my opinion, is an algebraic Bethe ansatz for medium range spin chains.

This work is very rigorous and I only identified minor questions and a few typos. I added some minor questions/suggestions in the "Requested changes" section below.

Finally, this is a very original paper, presenting a well structured and systematic approach to classify integrable models in restricted Hilbert spaces. It will certainly become an important reference to construct and investigate models of this type in the future. It attends all the criteria of SciPost Physics and I recommend it for publication once the comments below are addressed.

Requested changes

Questions/Comments:

  1. In the paragraph after equation (3) and in section 4.2, the authors talk about the GLL formulation, but no reference is provided. Is this formulation new? If not, can they please add a reference on the topic?

  2. In the paragraph about GLL, after equation (3), the authors mention "The G-operator contains the same information as the R-matrix, but acts on one less site." At this stage this looks confusing, since a standard R-matrix depends on only two sites. So, this sounds like "the G-operator depends on one site". The authors write a similar sentence later in the manuscript, after explaining the construction of medium range chains. There, it is very clear. I would suggest removing the sentence above from the Introduction.

  3. In the first paragraph on page 7, do the authors really mean $e^{Q_1} \sim U$, or would it be $e^{Q_1} \sim U^{r-1}$, where r is the range?

  4. In equation (25) and footnote 4, should be $\check{\mathcal{L}}$ instead of just $\mathcal{L}$?

  5. Is there any intuitive way to see that the dimension of the constrained Hilbert space grows as powers of the golden ratio (eq. (37))?

  6. I don't understand two of the steps in equation (61). The authors wrote "Since the commutation relations of $\Pi_{a_1a_2}$ and $\mathcal{L}_{a_1a_2j}(u)$ are the same as those of $\Pi_{a_1a_2}$ and $\mathcal{P}{a_1j} \mathcal{P}$" and then they write the equation. Since $\mathcal{L}_{a_1a_2j}$ only becomes $\mathcal{P}{a_1j} \mathcal{P}$ at $u=0$, I don't see why the commutation relations should be the same. Does this statement follow from any equations earlier in the paper? Can the authors please clarify? I also don't see why this can be used to justify the second equality.

  7. Is there a typo in the definition of $\Pi_{AB}$ in the first line of page 25? The index structure looks strange.

  8. The authors identify and analyse special points in equation (156-157). There is a choice that was not investigated: namely the case where $(1-7z^4+2z^6)=0$, which would make the term of type $P_1N_2P_3$ disappear. This happens for one imaginary point $z\sim 0.599926644682794 i$ and two real points: $z\sim 0.633814801916040$ and $z\sim 1.85962113772469$ (and their negative counterparts). Can the authors make any comments about these values of $z$? Is there any chance that phase transitions could happen also at these real points?

Typos:

  1. Just after equation (3), "Yang-Baxter..." -> "the Yang-Baxter..."
  2. In the paragraph before equation (4), "statistical physical models"-> "statistical physics models"
  3. In the last paragraph before section 2, "rasied" -> "raised"
  4. Just after equation (72), "...is contains two..." -> "...contains two..."
  5. Last line in page 15, "to proved" -> "to prove"
  6. Before equation (90), "congigurations" -> "configurations"
  7. Eqs. (156) and (157) should just be one equation (i.e. 156), right?

Recommendation

Publish (surpasses expectations and criteria for this Journal; among top 10%)

---

## Round 3 · Referee Report · Anonymous (Referee 2) · 2025-3-31

Strengths

  1. The paper provides the first systematic approach to construct integrable models in restricted Hilbert spaces.
  2. They constructed a new family of integrable models and provided a first investigation on their properties.
  3. The paper is well written and well structured, with a clear introduction and an interesting discussion presented in the conclusions. Appropriate references are also provided.

Weaknesses

None that I can see.

Report

The authors replied to all the points raised both by myself and Referee 1.

This is a very original and relevant paper. The changes further improved its clarity and I recommend it to publication in SciPost Physics in its present form.

Requested changes

None

Recommendation

Publish (surpasses expectations and criteria for this Journal; among top 10%)

---

## Round 3 · Author Response

We thank the reviewers for the questions and useful comments on our paper.

On the comments of reviewer 1: 1: We did not find many-body chiral symmetry for any of these operators, but it doesn’t mean that it’s not there. Certainly the models do not commute with C= prod_i sigma^z_i 2: We agree that sometimes it can be nice to explicitly see the algebraic equations resulting from the integrability condition, although it is not always helpful. We added an example in the text after equation (110). 3: The method on page 16 is analogous to the method described on page 8, with some adaptations due to the constrained Hilbert space which are discussed in section 4. 4: We kept one subsection to emphasise that there is only a single solution to the integrability condition in this case, and so the different integrable models at each range are clearly listed in the table of contents. 5: We are not aware of the definition of a “disordered” integrable model. If it is possible to derive such models from a solution of the Yang—Baxter equation then our method should apply, but we are not sure.

On the minor changes suggested by reviewer 1: 1: We feel the potential vagueness of the first sentence is clarified in the second sentence. 2: Algebraic origin refers to structures related to the Yang—Baxter equation, which is expanded upon in the second paragraph. 3: We agree there are many terms which are unfamiliar to non integrability experts. However, expanding upon each of these would disturb the flow of the introduction. For example, see “common algebraic framework for solving such models based on the Yang–Baxter equation has since been developed, known as the algebraic Bethe ansatz [5].” The only really important thing to know is that it is some algebraic method which allows for the solution of integrable models. The interested reader can then look at [5] for more details. 4: We joined the references 5: We fixed this 6: We mention in the caption that this dips in the plot correspond to the critical points, we think it is clear now.

On the questions/comments of reviewer 2: 1: The exact `GLL formulation’ was introduced in 2108.02053, we now mention this after equation (3). 2: In this paragraph we are speaking about higher range integrability. The GLL formulation does not make sense for range 2 models, since the Lax operator needs to act diagonally on the first on last site (so it would be trivial in this case). 3: Yes, we fixed this. 4. Yes, we fixed this. 5. One way is to set up a recursion relation for this dimension. On the open chain dim_L = dim_{L-1}+dim_{L-2}, which is solved by the Fibonacci numbers, whose asymptotics are given by powers of the golden ratio. This can be slightly modified on the periodic chain. 6. This follows from the discussion around equation (57) and (58). Since each operator in the checked Lax can be written in a form which commutes with Pi_A, we only need to pay attention to the commutation relations between Pi_A and the permutation operators in the Lax. 7. The indices are correct. First of all, we omitted checks from the R matrices between equations (74)-(77) which we now fixed. Applying the permutation operator P_{AB} leads to a slightly strange index structure on the projectors in the unchecked R matrix. 8. Indeed there are different values of z for which different operators in the double golden chain vanishes. Phase transitions typically occur when the ground state degeneracy of the system changes, this does not occur when the PNP term of the Hamiltonian vanishes. We don’t think these values of z will end up being special.

We thank reviewer 2 for pointing out the typos, which we fixed.

---

## Round 3 · List of Changes

• We added an example for the algebraic equations arising from the integrability condition after (110).
  • We joined the references in the introduction.
  • We now mention in the caption of figure 10 the locations of the critical points.
  • We added the reference for the origin of `GLL formulation’.
  • The first charge Q_1 for this models is actually a generalised momentum operator, such that exp(Q_1)=U^{r-1}. We fixed this.
  • Fixed confusion between L and \check{L}
  • Fixed numerous small typos

---

## Editorial Decision

published